# Enhancing Multifunctionality in Agricultural Landscapes with Native Woody Vegetation

**James Eggers** [1,*] , **Shannon Davis** [1,2], **Crile Doscher** [1] **and Pablo Gregorini** [2,3]

1   School of Landscape Architecture, Faculty of Environment, Society and Design, Lincoln University, Lincoln 7674, New Zealand
2   Centre of Excellence Designing Future Productive Landscapes, Lincoln University, Lincoln 7647, New Zealand
3   Department of Agricultural Science, Faculty of Agriculture and Life Sciences, Lincoln University, Lincoln 7674, New Zealand
*   Correspondence: james.eggers@lincoln.ac.nz

**Abstract:** The re-integration of native woody vegetation within agricultural areas has the potential to support multifunctional productive landscapes that enhance livestock welfare and restore habitat for native wildlife. As there is minimal research on this issue in Aotearoa New Zealand, this study aimed to identify species of native woody vegetation and propose spatial configurations and site designs to increase multifunctionality on a case study site. The three components of a multifunctional agricultural landscape focused on in this study were (1) enhancing foraging opportunities for livestock, (2) optimizing shade and shelter, and (3) establishing native bush bird habitat. During the first phase, sixty-three suitable species were identified and assigned scores based on the primary objectives and site constraints. This produced four optimized plant lists, one each for the three multifunctional components identified above and one combined multifunctional list incorporating those scores with additional environment and soil scores. The second phase used design thinking methodology to strategically locate these plants within an established case study site. Nine different planting configurations (three for each multifunctional component) were proposed and then, informed by site-specific opportunities and constraints, located on the case study site to produce three individual site designs. Finally, these three site designs were combined to propose an exemplar of a multifunctional agricultural landscape. The results indicate that reintegrating native woody vegetation has the potential to contribute toward multifunctional agricultural landscapes, proposing species and spatial layouts from which further investigation into livestock foraging, increased shade and shelter, and restoration of bush bird habitat can follow. This research advances sustainable land management practices by offering valuable insights into future agricultural landscape design.

**Keywords:** agricultural landscape; woody vegetation; forage; livestock; multifunctional landscape; native bush birds; shade; shelter

## 1. Introduction

Throughout much of early human history, the predominant source of food supply came from the hunter–gatherer lifestyle [1]. However, between 10,000–12,000 years ago, the development of agriculture by domesticating plants and animals allowed humans to settle in a fixed place [2,3]. However, as agricultural development continues globally, native vegetation has been and continues to be cleared at unprecedented rates, driving habitat fragmentation and biodiversity decline [4,5]. As the global population rises, the pressure to supply food continues to increase, resulting in the continued clearance of original vegetation to make space for food production [6]. While this process improves human food supply and nutrition, it contributes to global biodiversity loss [7,8], raising questions about the future of "modern" agricultural landscape approaches [1].

This trend is evident in agricultural landscapes such as the Canterbury Plains on Aotearoa New Zealand's South Island, where less than 0.5% of native vegetation remains [9],

and native bush birds are considerably outnumbered by introduced species [10]. To address the issue of biodiversity loss in dryland agricultural landscapes, this research investigates the reprioritization of local biodiversity on private farms [11,12]. Using plant lists developed for this study, alternative spatial strategies for optimal outcomes for livestock and native New Zealand bush birds were designed.

In Aotearoa New Zealand, there has been little research into the performance of native woody vegetation in agricultural landscapes; in fact, Tozer et al. (2021) recommend that "Additional research is required into the potential use of native woody plants in similar ways to those acknowledged and proposed for exotic species". However, that study focuses on hill country landscapes and may not directly apply to the east coast dryland agricultural landscapes investigated here. It does, however, acknowledge that the east coast is a target region for incorporating woody species into farming systems, particularly in areas with limited water availability. The study also highlights the potential benefits of trees and shrubs, such as enhanced foraging opportunities, shade, shelter, and additional habitat for bush birds once the vegetation becomes established [13].

While this study considers native woody vegetation, it also recognizes the importance of exotic vegetation within agricultural landscapes. As many exotic species used in Aotearoa New Zealand agricultural landscapes are deciduous (as opposed to most native woody vegetation being evergreen), leaf fall in winter increases soil organic matter and allows additional light to reach the ground, reducing the frequency and duration of frosts [14]. Furthermore, exotic vegetation tends to grow taller more rapidly when compared to native vegetation, enabling faster establishment of shade and shelter, one of the aims of this study (M. Bloomberg, personal communication, 25 July 2022) [15].

Native vegetation also provides multiple food sources for native bush birds. Agricultural landscapes in the Canterbury region of Aotearoa New Zealand are dominated by exotic gymnosperms, where the seeds are located within cones. These are often difficult to access for native bush birds, which have evolved to gain food primarily from flowering angiosperms. An increase in native flowering plants would provide a food supply for native bush birds, encouraging the repopulation of these agricultural landscapes [16,17].

### 1.1. Literature Review

This section reviews the literature on the three components of multifunctional landscapes considered in this study: livestock foraging, shade and shelter, and native bush bird habitat. It also introduces landscape ecology theory and its relevance to the research, concluding with the study's objectives.

### 1.1.1. Livestock Foraging

The first component considered in this research focuses on enhancing foraging opportunities for livestock within dryland agricultural landscapes. This study categorizes livestock as sheep and cattle. Foraging is a term used to describe the process where livestock seek food to consume to enhance their overall nutrient and energy intake [18–20]. Optimal foraging theory assumes livestock carry out foraging activities that maximize energy input and minimize energy output. As food preference in livestock is not random, time and energy are expended searching for forage [21,22].

The use of woody vegetation as forage has long been recognized [23–28]. Shrubs are used by farmers to lengthen the grazing season and can better tolerate poor soils and drought conditions where other vegetation finds growth difficult [29]. However, it is common within modern agricultural systems for livestock to rely exclusively on grass-based diets, even though they may not provide the range of nutrients required for optimal health. Increasing vegetation diversity within a farming system improves animal welfare as livestock can exhibit natural foraging behaviors and seek out those plant species which optimize their health [30,31].

The spatio-temporal arrangement of forage species also affects their consumption. When offered the choice between diverse forage mixtures and spatially adjacent monocultures, performance was enhanced by the consumption of the diverse mixture [32,33]. The spatial distribution of forage plants and the temporal distribution of foraging trips are strongly integrated and are more efficient within distances of 40 m [34]. Furthermore, the availability of shade in areas with high-quality forage increases overall consumption [35]. This will guide the location of forage vegetation when applied to the case study site.

### 1.1.2. Shade and Shelter

The second component considered in this research focused on providing shade and shelter within dryland agricultural landscapes. The New Zealand Animal Welfare Act 1999 [36] states that livestock should be protected from extreme weather events such as heat, cold winds, and heavy precipitation. Shade and shelter in agricultural landscapes provide this protection by mitigating climatic extremes [37], allowing regulation of body heat and stress associated with extreme heat and cold [38]. According to Farm Forestry New Zealand (2011) [39], the optimum porosity of shelter plantings should be between 40–60%, which can be determined by species composition and overall width [40,41], with shelter extending downwind approximately 2.5 times the height of the trees [42]. Furthermore, to reduce the amount of wind moving around the ends of the shelter vegetation, the overall length should be at least ten times the height [43].

Common shelter features in Aotearoa New Zealand include vegetated shelterbelts [44], rows of woody vegetation that reduce windspeeds and modify the microclimate [45]. While some distinguish between windbreaks and shelterbelts [46], the term is often used interchangeably and also with terms such as hedge, hedgerow, vegetative barrier, windbreak, or wind barrier [47,48]. Canopy trees are an effective method for providing shade in agricultural landscapes. The shade provided depends on the canopy's orientation and density of leaves and branches, which can be controlled by species selection [38].

### 1.1.3. Native Bush Birds

Finally, the third component considered in this research is habitat provision for native bush birds. It is widely accepted that native birds are an indicator species for wider environmental health [49–51]. Therefore, it can be assumed that areas with abundant populations may be in good health and vice versa. The primary cause of native bird loss is the removal and fragmentation of habitat [52,53]. Combined with increased competition for resources and predation [54–56], many bush bird populations have become locally extinct, contributing to an overall degradation of ecosystem health within agricultural landscapes [52,57]. Therefore, restoring native bush is critical to increasing the numbers of native bush birds in agricultural landscapes and restoring overall ecosystem health [58,59].

The wider landscape surrounding the case study site is the eastern Selwyn District (an area bounded by Christchurch City to the northeast, State Highway 1 to the northwest, the Rakaia River to the southwest and the Pacific Ocean to the southeast). Native bush birds previously seen in this area are fantail/pīwakawaka (*Rhipidura fuliginosa*), bellbird/korimako (*Anthornis melanura*), South Island robin/kakaruai (*Petroica australis*), tomtit/miromiro (*Petroica macrocephala*), silvereye/tauhou (*Zosterops lateralis*), New Zealand pigeon/kererū (*Hemiphaga novaeseelandiae*), welcome swallow/warou (*Hirundo neoxena*), grey warbler/riroriro (*Gerygone igata*), rifleman/tītitipounamu (*Acanthisitta chloris*), sacred kingfisher/kōtare (*Todiramphus sanctus*), brown creeper/pīpipi (*Mohoua novaeseelandiae*), pipit/pīhoihoi (*Anthus novaeseelandiae*), morepork/ruru (*Ninox novaeseelandiae*), tūī/parson bird (*Prosthemadera novaeseelandiae*), long-tailed cuckoo/koekoeā (*Urodynamis taitensis*), and shining cuckoo/pīpīwharauroa (*Chrysococcyx lucidus*) [60].

### 1.1.4. Landscape Ecology

A key objective of this study was to determine the optimal location for native bush bird habitat in the case study site. As landscape ecology theory provides tools for spatial planning, focusing on the interaction of spatial patterns and ecological processes [61–63], it was used to identify this location. By establishing patches and corridors of woody vegetation, landscape ecology improves connectivity in fragmented ecological networks, supporting the return of native bird species to agricultural landscapes and increasing their chances of survival [61,64]. Meurk and Hall (2006) [64] propose patch and corridor configurations for spatial connectivity, including 6.25 ha core sanctuaries spaced ca. 5 km apart, 1.6 ha patches 1–2 km apart for insectivorous and frugivorous bush birds, and 0.01 ha patches 0.2 km apart for finer-grained stepping-stones. Given that climatic edge effects reach ca. 50 m into forest remnants [65,66], a 6.25 ha patch would have a core area of ca 1.8 ha, surrounded by a 4.45 ha buffer zone, the minimum size to support sensitive bush bird species.

The research objectives of this study are to first: identify species of native woody vegetation that are most suited for enhancing three components of a multifunctional dryland agricultural landscape: (1) enhanced foraging opportunity for livestock, (2) increased shade and shelter, (3) restoration of native bush bird habitat. Second, to propose spatial configurations that integrate these species to improve landscape multifunctionality, the study uses a place-based design approach for the case study site of Ashley Dene Farm Canterbury Plains on Aotearoa New Zealand's South Island. As there has been no published research on the integration of Aotearoa New Zealand native plants into agricultural environments to investigate their potential for livestock forage, shade and shelter, and bird habitat, this study represents a significant knowledge contribution to both Aotearoa New Zealand and the global research community.

## 2. Methods

### 2.1. Overview

Figure 1 below outlines the two-stage methodology used in this study. Firstly, plant lists were developed, and then the 20 highest-ranking species from each list were used to create nine planting configurations (three for each component of a multifunctional dryland agricultural landscape). These configurations were then placed in suitable locations on the case study site to produce a site design for each component of a multifunctional dryland agricultural landscape. Finally, the three site designs were integrated with the combined multifunctional plant list to form a combined multifunctional design.

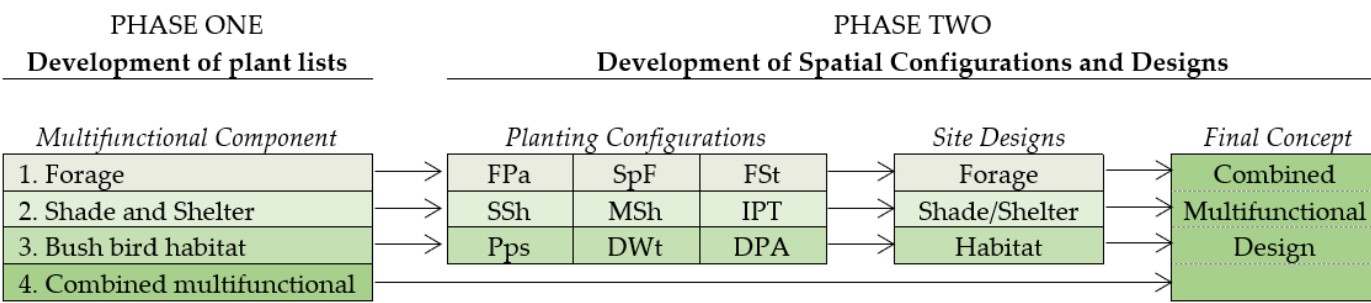

**Figure 1.** Outline of research methodology: "Phase One", the development of the plant lists focusing on three components of a multifunctional dryland agricultural landscape, which are then applied to "Phase Two", the development of spatial configurations and designs. Key: FPa: Forage patches; SpF: Spreading forage; FSt: Forage strips; SSh: Straight shelter; MSh: Meandering shelter; IPT: In-paddock shade trees; Pps: Area of bush bird habitat focused on existing poplar trees; DWt: Area of bush bird habitat focused on existing drainage/wetland corridor; and DPA: A densely planted area of the bush bird habitat.

*2.2. Methodology: Phase One*

Developing the Plant Lists

Four plant lists (see Table 1) were created by scoring 63 species of native woody vegetation. These are listed below. With limited information on livestock preferences for native vegetation, it was assumed that all herbivores, including domestic and wild species, shared similar preferences to sheep and cattle.

1. Forage characteristics: palatability, tolerance to defoliation, and growth rates.
2. Shade and shelter characteristics: shade provision, plant height, canopy size, and shelter provision.
3. Native bush bird habitat characteristics: availability of food sources and quality of nesting sites.
4. A combined multifunctional list with scores from lists 1–3 and scores for environmental tolerances of frost, drought, wind, sun/shade preferences, soil preferences of drainage, depth, moisture, and type, and conservation threat status [67].

Scores between 1–5 (except conservation status 0–8) were assigned to each characteristic based on suitability for use in dryland agricultural landscapes.

Weightings between ×0.25 and ×1.5 were assigned to each characteristic based on a subjective evaluation of the importance within this study. These weightings may be adjusted to suit the requirements of each case study.

Weighted scores for each of the three multifunctional components were combined to identify which species could best provide (1) forage, (2) shade and shelter, and (3) native bush bird habitat. The 20 highest-scoring species were used for the planting configurations and site designs from Figure 1 above. Following this, scores for environment and soil were combined with those for each of the three multifunctional components to develop the (4) multifunctional plant list. This list was applied to the combined multifunctional design.

**Table 1.** Characteristics, scores and weightings assigned to species of native woody vegetation with potential use as multifunctional components in dryland agricultural landscapes.

| Species of Native Woody Vegetation | Forage | | | | Shade and Shelter | | | | | Habitat | | | Environment and Soil | | | | | | | | | | Combined Lists | |
|---|---|---|---|---|---|---|---|---|---|---|---|---|---|---|---|---|---|---|---|---|---|---|---|---|
| | Palatability | Growth Rate | Defoliation | Total Forage Score | Shade Provision | Plant Height | Canopy Size | Shelter Provision | Total Shade and Shelter Score | Food Sources | Nesting Sites | Total Bush Bird Habitat Score | Conservation Status | Frost Tolerance | Drought Tolerance | Wind Tolerance | Sun and Shade | Soil Drainage | Soil Depth | Soil Moisture | Soil Type | Total Environment and Soil Score | Total Combined Multifunctional Score | Total Combined Multifunctional Scores (Weighted). (Score Used in Plant Lists) |
| Sp.1 | 1–5 | 1–5 | 1–5 | #/15 | 1–5 | 1–5 | 1–5 | 1–5 | #/20 | 1–5 | 1–5 | #/10 | 0–8 | 1–5 | 1–5 | 1–5 | 1–5 | 1–5 | 1–5 | 1–5 | 1–5 | #/48 | #/93 | #/95.75 |
| Sp.2 | 1–5 | 1–5 | 1–5 | #/15 | 1–5 | 1–5 | 1–5 | 1–5 | #/20 | 1–5 | 1–5 | #/10 | 0–8 | 1–5 | 1–5 | 1–5 | 1–5 | 1–5 | 1–5 | 1–5 | 1–5 | #/48 | #/93 | #/95.75 |
| Sp.3 | 1–5 | 1–5 | 1–5 | #/15 | 1–5 | 1–5 | 1–5 | 1–5 | #/20 | 1–5 | 1–5 | #/10 | 0–8 | 1–5 | 1–5 | 1–5 | 1–5 | 1–5 | 1–5 | 1–5 | 1–5 | #/48 | #/93 | #/95.75 |
| … | | | | … | | | | | … | | | … | | | | | | | | | | … | … | … |
| Sp.63 | 1–5 | 1–5 | 1–5 | #/15 | 1–5 | 1–5 | 1–5 | 1–5 | #/20 | 1–5 | 1–5 | #/10 | 0–8 | 1–5 | 1–5 | 1–5 | 1–5 | 1–5 | 1–5 | 1–5 | 1–5 | #/48 | #/93 | #/95.75 |
| Weighting and scores | × 1 | × 1 | × 1 | #/15 | × 1.5 | × 1 | × 1 | × 1.5 | #/25 | × 0.5 | × 1 | #/7.5 | × 0.25 | × 1 | × 1.5 | × 1 | × 1 | × 1.5 | × 1.5 | × 1.5 | × 0.25 | #/48.25 | | |

Key: #/15 total forage scores out of 15; #/15: weighted total forage scores out of 15; #20 total shade and shelter scores out of 20; #/25: weighted total shade and shelter scores out of 25; #10 total bush bird habitat scores out of 10; #/7.5: weighted total bush bird habitat scores out of 7.5; #48 total environment and soil scores out of 48; #/48.25: weighted total environment and soil scores out of 48.25; #93 total combined scores out of 93; #/95.75: weighted total combined scores out of 95.75.

*2.3. Methodology: Phase Two*
Design Thinking Methodology

Design thinking methodology (Figure 2) was used to develop planting configurations and site designs. The use of this method is relatively new in agricultural landscapes, enabling unique perspectives on design issues [68]. The "Empathize" step involved discussions with agricultural and ecological experts and a comprehensive site inventory of climate, vegetation, soils, water, built infrastructure, forage, shade, shelter, bush bird habitat, and site limitations. A broader-scale inventory of vegetation provided insights into the wider ecological network. The subsequent "Define" stage identified key issues and aspirations for integrating native woody vegetation. In the "Ideate" stage, nine spatial planting configurations were designed that aligned with the site inventory and addressed the identified issues. This was followed by developing three "Prototypes", or site designs, representing multifunctional components of forage, shade and shelter, and bush bird habitat. Finally, a comprehensive "Prototype" combining all three components was created for the multifunctional design. This was then tested using a SWOT analysis to refine the prototypes and inform further design iterations.

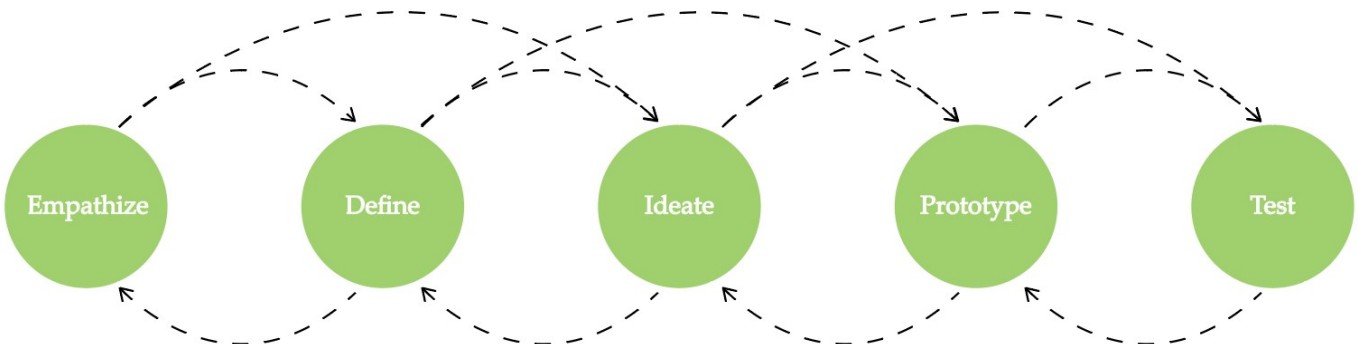

**Figure 2.** The five steps of the design thinking methodology, adapted from Stanford Design School.

*2.4. The Case Study Site: Ashley Dene Farm*

A case study site of Lincoln University-owned Ashley Dene farm on the Canterbury Plains of Aotearoa New Zealand, was selected to investigate and test spatial designs of native vegetation in agricultural landscapes (see Figure 3 below).

Ashley Dene dryland farm reflects the wider Canterbury Plains landscape, where less than 0.5% of the original vegetation remains [9]. Approximately 700 years ago, low, shrubby kānuka forests dominated the landscape, including matagouri/tūmatakuru (*Discaria toumatou*), small-leaved kōwhai (*Sophora microphylla*) and tussock spp. on the shallow soils and tall podocarp forests with kahikatea (*Dacrycarpus dacrydioides*), tōtara (*Podocarpus totara*), mataī (*Prumnopitys taxifolia*), mānuka (*Leptospermum scoparium*), harakeke (*Phormium tenax*), raupō (*Typha orientalis*) and rushes on the deeper soils [69,70]. Today, the landscape is dominated by exotic *Populus* spp., *Eucalyptus* spp., radiata pine, macrocarpa shelterbelts and gorse hedges (*Ulex europaeus*).

Livestock at Ashley Dene dryland farm as of February 2023 includes sheep and cattle species. Sheep numbers are 610 mixed-age Coopworth ewes, 70 mixed-age Romney ewes, 70 mixed-age Batten ewes, 170 Coopworth/Romney two-tooth ewes, 667 mixed-sex lambs, 22 two-tooth rams, and 5 mixed-age rams, and cattle numbers are 40 rising 3-year-old Angus cows, 40 weaner calves and 1 rising 2-year-old bull (A. Greer, personal communication, 25 June 2023).

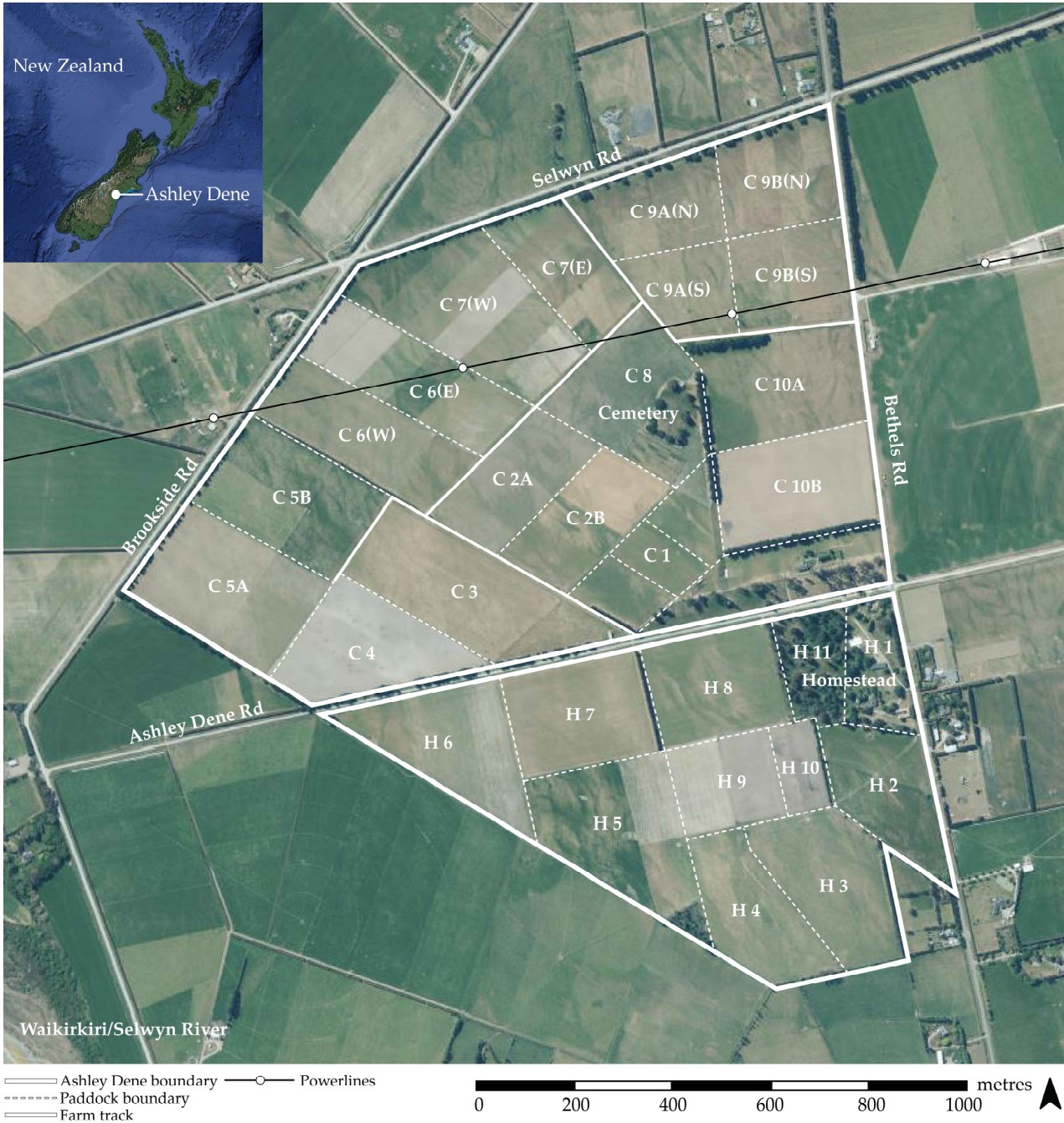

**Figure 3.** Ashley Dene farm, showing key features, paddock numbers and high-voltage power lines. Design does not apply to H1 and H11 as these are the primary workshop and accommodation areas.

### 2.5. Phase Two: Planting Configurations

The top 20 native woody vegetation species were selected from plant lists 1–3 to create planting configurations for each function. Existing woody vegetation layouts in agricultural landscapes informed these configurations.

### 2.6. Phase Two: Site Designs

Three site designs were created using the case study method [71], placing each planting configuration in the most suitable location based on environment and soil characteristics and site inventory from Figure 3. Each design maximized the benefits of a specific multi-functional landscape component, with minimal consideration for other functions.

*2.7. Phase Two: Combined Multifunctional Design*

The three site designs were integrated to create a combined multifunctional design for the case study site, prioritizing landscape multifunctionality while minimizing costs and disruption to existing farm infrastructure. Preference was given to bush bird habitat when deciding the spatial arrangement of woody vegetation, as native bush birds have specific habitat requirements in agricultural landscapes. The accuracy of plant placement on the site was improved using the combined multifunctional plant list (plant list 4).

## 3. Results

This section presents the plant lists, spatial planting configurations, site designs for each multifunctional landscape component and the combined multifunctional site design.

*3.1. Phase One: Plant Lists*

This section presents the top 20 highest-scoring plants for the three multifunctional components in this study and the combined multifunctional list. A complete list is available in Appendix A.

3.1.1. Enhancing Foraging Opportunities in Dryland Agricultural Landscapes

Table 2 below shows the top 20 rated species for enhancing foraging opportunities for livestock by combining palatability, tolerance to defoliation, and plant growth rates.

**Table 2.** Top 20 native woody vegetation species for foraging (weighted scores).

| Botanical Name | Common Name | Palatability | Defoliation | Growth Rate | Total |
|---|---|---|---|---|---|
| *Aristotelia serrata* | makomako, wineberry | 5.0 | 4.0 | 3.0 | 12.0 |
| *Coprosma robusta* | karamū | 5.0 | 3.0 | 4.0 | 12.0 |
| *Teucridium parvifolium* | teucridium | 3.0 | 4.0 | 5.0 | 12.0 |
| *Pittosporum eugenioides* | lemonwood, tarata | 3.0 | 3.0 | 5.0 | 11.0 |
| *Pittosporum tenuifolium* | black matipo, kōhūhū | 3.0 | 3.0 | 5.0 | 11.0 |
| *Griselinia littoralis* | broadleaf, kāpuka | 5.0 | 3.0 | 3.0 | 11.0 |
| *Cordyline australis* | cabbage tree, ti kōuka | 3.0 | 5.0 | 3.0 | 11.0 |
| *Pseudopanax arboreus* | five-finger, whauwhaupaku | 5.0 | 3.0 | 3.0 | 11.0 |
| *Veronica salicifolia* | koromiko | 5.0 | 1.0 | 5.0 | 11.0 |
| *Coprosma lucida* | karangu, shining karamū | 5.0 | 3.0 | 3.0 | 11.0 |
| *Veronica strictissima* | Banks Peninsula hebe | 5.0 | 1.0 | 5.0 | 11.0 |
| *Coprosma virescens* | mikimiki | 3.0 | 3.0 | 5.0 | 11.0 |
| *Aristotelia fruticosa* | mountain wineberry | 5.0 | 4.0 | 2.0 | 11.0 |
| *Hoheria angustifolia* | narrow-leaved houhere | 3.0 | 4.0 | 3.0 | 10.0 |
| *Pennantia corymbosa* | kaikōmako | 4.0 | 3.0 | 3.0 | 10.0 |
| *Carpodetus serratus* | marbleleaf, putaputāwētā | 3.0 | 4.0 | 3.0 | 10.0 |
| *Coprosma linariifolia* | yellow wood | 3.0 | 4.0 | 3.0 | 10.0 |
| *Coprosma pedicellata* | mikimiki | 3.0 | 4.0 | 3.0 | 10.0 |
| *Olearia bullata* | shrub daisy | 3.0 | 4.0 | 3.0 | 10.0 |
| *Coprosma propinqua* | mikimiki | 3.0 | 3.0 | 4.0 | 10.0 |

### 3.1.2. Increasing Shade and Shelter in Dryland Agricultural Landscapes

Table 3 below shows the top 20 rated species for increasing shade and shelter by combining shade, height, canopy, and shelter scores.

**Table 3.** Top 20 native woody vegetation species for increasing shade and shelter (weighted scores).

| Botanical Name | Common Name | Shade | Height | Canopy | Shelter | Total |
|---|---|---|---|---|---|---|
| *Podocarpus totara* | tōtara | 7.5 | 5.0 | 5.0 | 7.5 | 25.0 |
| *Prumnopitys taxifolia* | mataī, black pine | 7.5 | 5.0 | 5.0 | 7.5 | 25.0 |
| *Dacrycarpus dacrydioides* | kahikatea | 7.5 | 5.0 | 4.0 | 7.5 | 24.0 |
| *Elaeocarpus hookerianus* | pōkākā | 7.5 | 4.0 | 5.0 | 6.0 | 22.5 |
| *Pittosporum eugenioides* | lemonwood, tarata | 6.0 | 4.0 | 4.0 | 7.5 | 21.5 |
| *Hoheria angustifolia* | narrow-leaved houhere | 7.5 | 4.0 | 3.0 | 6.0 | 20.5 |
| *Plagianthus regius* | lowland ribbonwood | 7.5 | 4.0 | 3.0 | 6.0 | 20.5 |
| *Dodonaea viscosa* | akeake | 6.0 | 3.0 | 3.0 | 6.0 | 18.0 |
| *Pittosporum tenuifolium* | black matipo, kōhūhū | 6.0 | 3.0 | 3.0 | 6.0 | 18.0 |
| *Corynocarpus laevigatus* | karaka | 4.5 | 4.0 | 3.0 | 6.0 | 17.5 |
| *Griselinia littoralis* | broadleaf, kāpuka | 6.0 | 2.0 | 5.0 | 4.5 | 17.5 |
| *Cordyline australis* | cabbage tree, ti kōuka | 3.0 | 4.0 | 2.0 | 6.0 | 15.0 |
| *Pseudopanax arboreus* | five-finger, whauwhaupaku | 4.5 | 3.0 | 3.0 | 4.5 | 15.0 |
| *Aristotelia serrata* | makomako, wineberry | 3.0 | 4.0 | 3.0 | 4.5 | 14.5 |
| *Pennantia corymbosa* | kaikōmako | 3.0 | 4.0 | 3.0 | 4.5 | 14.5 |
| *Olearia fragrantissima* | fragrant tree daisy | 4.5 | 3.0 | 2.0 | 4.5 | 14.0 |
| *Kunzea ericoides* | kānuka | 3.0 | 3.0 | 3.0 | 4.5 | 13.5 |
| *Carpodetus serratus* | marbleleaf, putaputāwētā | 1.5 | 4.0 | 3.0 | 4.5 | 13.0 |
| *Pseudopanax crassifolius* | lancewood, horoeka | 1.5 | 4.0 | 3.0 | 4.5 | 13.0 |
| *Leptospermum scoparium* | mānuka | 3.0 | 2.0 | 3.0 | 4.5 | 12.5 |

### 3.1.3. Restoring Native Bush Bird Habitat in Dryland Agricultural Landscapes

Table 4 below shows the top 20 rated species for restoring native bush bird habitat by combining the number of food sources and quality of nesting sites.

**Table 4.** Top 20 native woody vegetation species for restoring native bush bird habitat (weighted scores).

| Botanical Name | Common Name | Food | Nesting | Total |
|---|---|---|---|---|
| *Pseudopanax arboreus* | five-finger, whauwhaupaku | 2.5 | 5.0 | 7.5 |
| *Sophora microphylla* | kōwhai, small-leaved kōwhai | 2.5 | 5.0 | 7.5 |
| *Cordyline australis* | cabbage tree, ti kōuka | 2.0 | 5.0 | 7.0 |
| *Carpodetus serratus* | marbleleaf, putaputāwētā | 2.0 | 5.0 | 7.0 |
| *Kunzea ericoides* | kānuka | 2.0 | 5.0 | 7.0 |
| *Hoheria angustifolia* | narrow-leaved houhere | 1.5 | 5.0 | 6.5 |
| *Elaeocarpus hookerianus* | pōkākā | 1.5 | 5.0 | 6.5 |
| *Prumnopitys taxifolia* | mataī, black pine | 1.5 | 5.0 | 6.5 |
| *Dacrycarpus dacrydioides* | kahikatea | 1.5 | 5.0 | 6.5 |
| *Corynocarpus laevigatus* | karaka | 1.0 | 5.0 | 6.0 |
| *Plagianthus regius* | lowland ribbonwood | 1.0 | 5.0 | 6.0 |

**Table 4.** *Cont.*

| Botanical Name | Common Name | Food | Nesting | Total |
|---|---|---|---|---|
| *Podocarpus totara* | tōtara | 1.0 | 5.0 | 6.0 |
| *Griselinia littoralis* | broadleaf, kāpuka | 2.5 | 3.0 | 5.5 |
| *Pittosporum eugenioides* | lemonwood, tarata | 2.0 | 3.0 | 5.0 |
| *Pittosporum tenuifolium* | black matipo, kōhūhū | 2.0 | 3.0 | 5.0 |
| *Aristotelia serrata* | makomako, wineberry | 1.5 | 3.0 | 4.5 |
| *Dodonaea viscosa* | akeake | 1.5 | 3.0 | 4.5 |
| *Olearia fragrantissima* | fragrant tree daisy | 1.5 | 3.0 | 4.5 |
| *Olearia paniculata* | akiraho, golden akeake | 1.5 | 3.0 | 4.5 |
| *Olearia avicenniifolia* | mountain akeake | 1.5 | 3.0 | 4.5 |

### 3.1.4. Enhancing Multifunctionality in Dryland Agricultural Landscapes

Table 5 below shows the top 20 rated species for creating a multifunctional agricultural landscape by combining forage, shade and shelter, native bush bird habitat, environmental tolerances, and soil preferences.

**Table 5.** Top 20 species of native woody vegetation (weighted scores) used for the combined multifunctional site design (groupings).

| Botanical Name | Common Name | Fo | SS | HT | ET | So | Total |
|---|---|---|---|---|---|---|---|
| *Podocarpus totara* | tōtara | 5.00 | 25.00 | 6.00 | 22.50 | 20.75 | 79.25 |
| *Hoheria angustifolia* | narrow-leaved houhere | 10.00 | 20.50 | 6.50 | 21.00 | 20.75 | 78.75 |
| *Elaeocarpus hookerianus* | pōkākā | 8.00 | 22.50 | 6.50 | 17.50 | 21.75 | 76.25 |
| *Plagianthus regius* | lowland ribbonwood | 8.00 | 20.50 | 6.00 | 19.50 | 20.75 | 74.75 |
| *Prumnopitys taxifolia* | mataī, black pine | 5.00 | 25.00 | 6.50 | 15.50 | 20.75 | 72.75 |
| *Cordyline australis* | cabbage tree, ti kōuka | 11.00 | 15.00 | 7.00 | 21.50 | 17.75 | 72.25 |
| *Pittosporum tenuifolium* | black matipo, kōhūhū | 11.00 | 18.00 | 5.00 | 19.00 | 18.50 | 71.50 |
| *Griselinia littoralis* | broadleaf, kāpuka | 11.00 | 17.50 | 5.50 | 20.50 | 17.00 | 71.50 |
| *Kunzea ericoides* | kānuka | 7.00 | 13.50 | 7.00 | 23.75 | 20.00 | 71.25 |
| *Dodonaea viscosa* | akeake | 9.00 | 18.00 | 4.50 | 19.50 | 18.75 | 69.75 |
| *Pittosporum eugenioides* | lemonwood, tarata | 11.00 | 21.50 | 5.00 | 14.50 | 15.50 | 67.50 |
| *Dacrycarpus dacrydioides* | kahikatea | 5.00 | 24.00 | 6.50 | 14.50 | 17.25 | 67.25 |
| *Olearia fragrantissima* | fragrant tree daisy | 9.00 | 14.00 | 4.50 | 18.00 | 20.75 | 66.25 |
| *Corynocarpus laevigatus* | karaka | 8.00 | 17.50 | 5.00 | 15.50 | 19.25 | 65.25 |
| *Olearia avicenniifolia* | mountain akeake | 9.00 | 10.00 | 4.50 | 22.50 | 18.50 | 64.50 |
| *Pseudopanax arboreus* | five-finger, whauwhaupaku | 11.00 | 15.00 | 7.50 | 14.50 | 16.25 | 64.25 |
| *Olearia paniculata* | akiraho, golden akeake | 9.00 | 12.50 | 4.50 | 19.00 | 19.25 | 64.25 |
| *Discaria toumatou* | matagouri, tūmatakuru | 9.00 | 10.00 | 2.00 | 23.50 | 19.00 | 63.50 |
| *Sophora microphylla* | kōwhai, small-leaved kōwhai | 6.00 | 11.00 | 6.50 | 21.50 | 18.50 | 63.50 |
| *Leptospermum scoparium* | mānuka | 7.00 | 12.50 | 4.00 | 23.50 | 16.25 | 63.25 |

Key: Fo: Forage; SS: Shade and shelter; HT: Native bush bird habitat; ET: Environmental tolerances; So: Soil preferences.

### 3.2. Phase Two: Planting Configurations and Site Designs

The following section presents the planting configurations and site designs for the three multifunctional components considered in the study.

### 3.2.1. Designing to Enhance Foraging Opportunity

Forage species were arranged in three spatial configurations: "Forage patches", "Spreading forage", and "Forage strips".

#### Forage Patches

Forage patches aim to provide various-sized patches and species compositions within individual paddocks. Taller forage species were fenced off in the center of the patch to provide forage for taller livestock, while shade-tolerant species were placed underneath, with sun-tolerant species extending out into the paddock. This allows livestock to select a diet according to their preferences. The fenced-off areas also allowed for natural regeneration by protecting new seedlings. Plant combinations varied between patches to increase dietary diversity (see Figure 4).

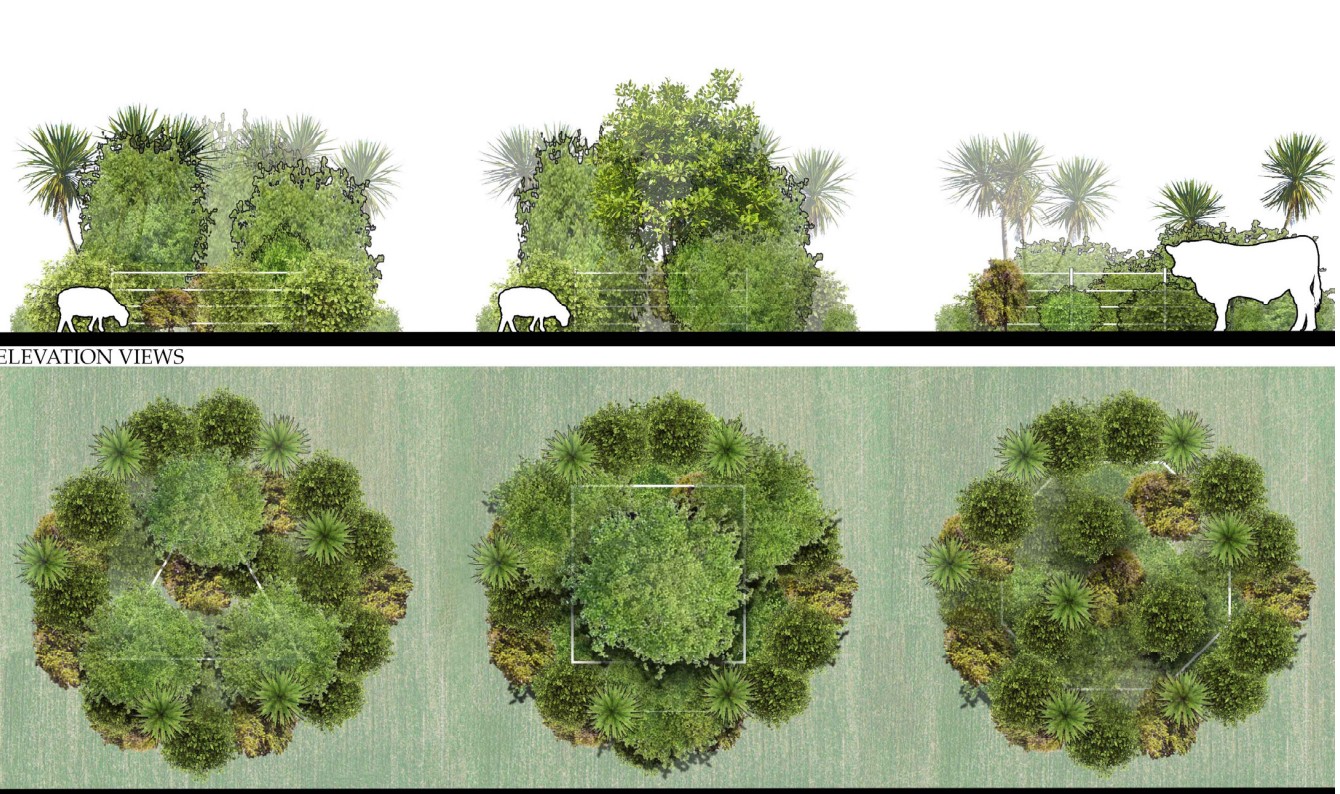

**Figure 4.** Elevation views (**top**) and plan views (**bottom**) of a forage patch configuration. The permanently fenced area in the center protects vegetation, allowing it to regenerate naturally.

#### Spreading Forage

The spreading forage configuration balances access to forage and prevents overbrowsing. Taller forage species are fenced off, with shade-tolerant species underneath and sun-tolerant species spreading into adjacent paddocks. This design allows controlled access to forage by moving livestock between paddocks, allowing plants to regrow. The configuration provides a more natural form for livestock, improving welfare with shade and shelter throughout the day and under different weather conditions. The fenced areas protect new seedlings, allowing for natural regeneration. Plantings replicate natural variations, enabling livestock to exhibit natural foraging behaviors (see Figure 5).

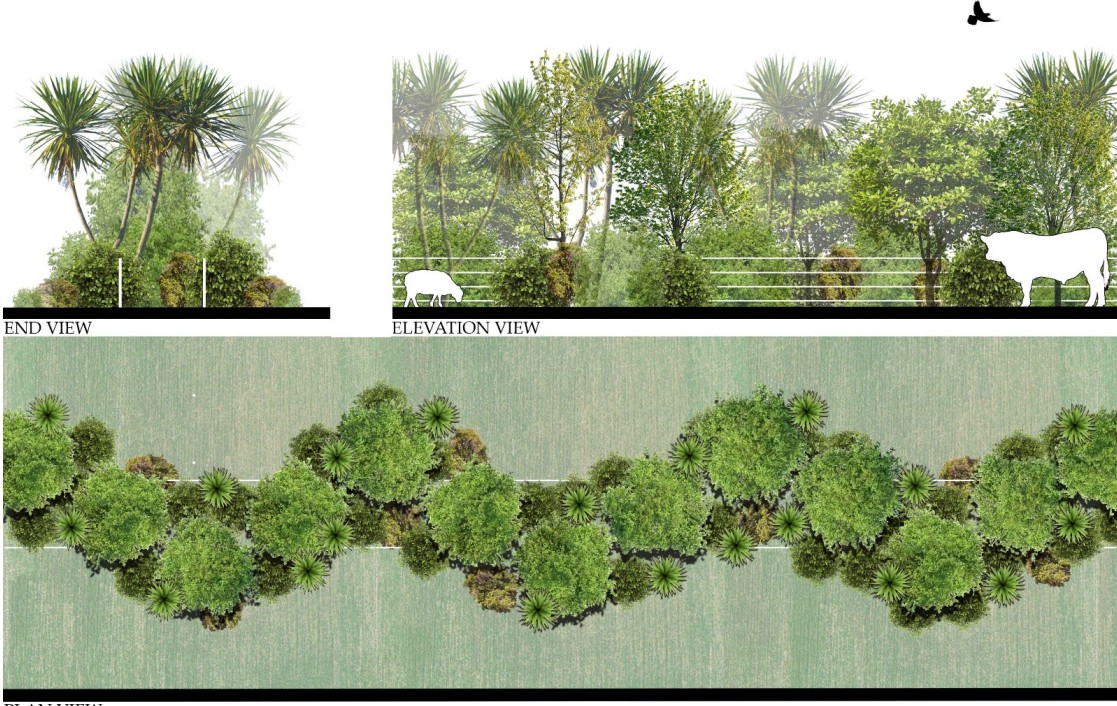

**Figure 5.** End and elevation views (**top**) and plan view (**bottom**) of the spreading forage configuration with two permanent fence lines running lengthways through the center.

Forage Strips

Forage strips maximize livestock foraging opportunities and are compatible with conventional farming systems that utilize farm machinery. These strips are within individual paddocks, with temporary fencing designating break-feeding areas. The design of the plantings replicates natural variations, enabling livestock to exhibit their natural foraging behaviors and choose their preferred diet. Taller forage species are positioned in the center of the strips, while lower-growing species remain underneath to optimize grazing space (see Figure 6).

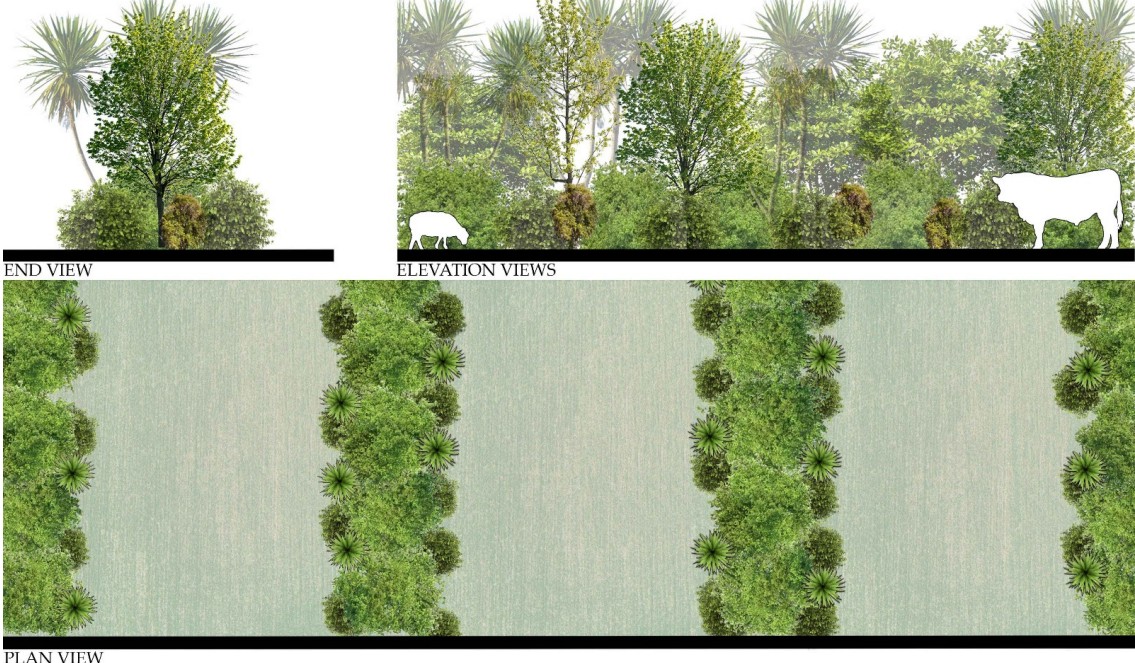

**Figure 6.** End and elevation views (**top**) and plan view (**bottom**) of the forage strip configuration.

### 3.2.2. Livestock Forage Site Design

Forage patches were placed in three locations: C8 and in C6(W) and H2 for lambing and calving. The number of forage species was limited to eight in lambing areas and increased in calving areas, following Wang et al. (2010) [72]. Taller plants were placed in potential calving areas for larger overall consumption and higher reach.

Spreading forage was placed in the center of the Cemetery Block and the center and south of the Homestead Block to balance forage availability while preserving grazing and cropping space.

Forage strips were implemented in five areas: C10A and C10B paddocks for maximum sun exposure throughout the day, C3 and C4 paddocks with northwest to southeast orientation, C5 paddock with northeast to southwest orientation, H6 paddock with north to south orientation, and H8 paddock with east to west orientation. Permanent fencing will replace temporary electric/virtual fencing (see Figure 7).

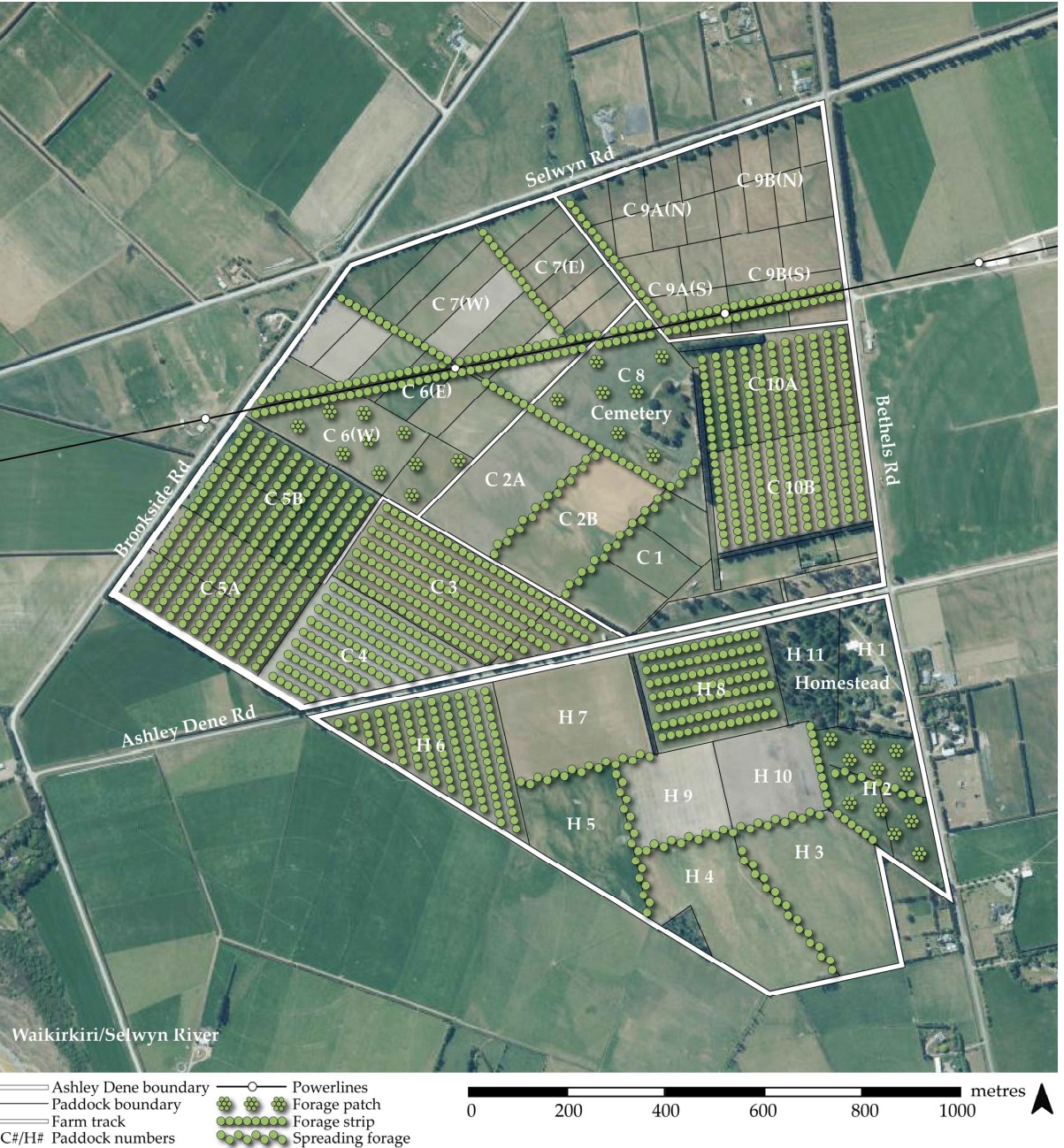

**Figure 7.** Livestock forage site design with proposed locations of each forage configuration.

### 3.2.3. Designing for Increased Shade and Shelter

Shade and shelter plantings were organized into three spatial configurations: "straight shelter", "meandering shelter", and "in-paddock shade trees". While livestock has a degree of access to forage plantings, access to shade and shelter plantings will be restricted. This allows the plantings to grow without the pressure of livestock browsing.

### Straight Shelter

This configuration was designed to provide shelter in areas with limited vegetation space, with taller growing species in the center and medium to low-growing species underneath. Lower-growing native species can also be planted beneath existing exotic shelterbelts (see Figure 8).

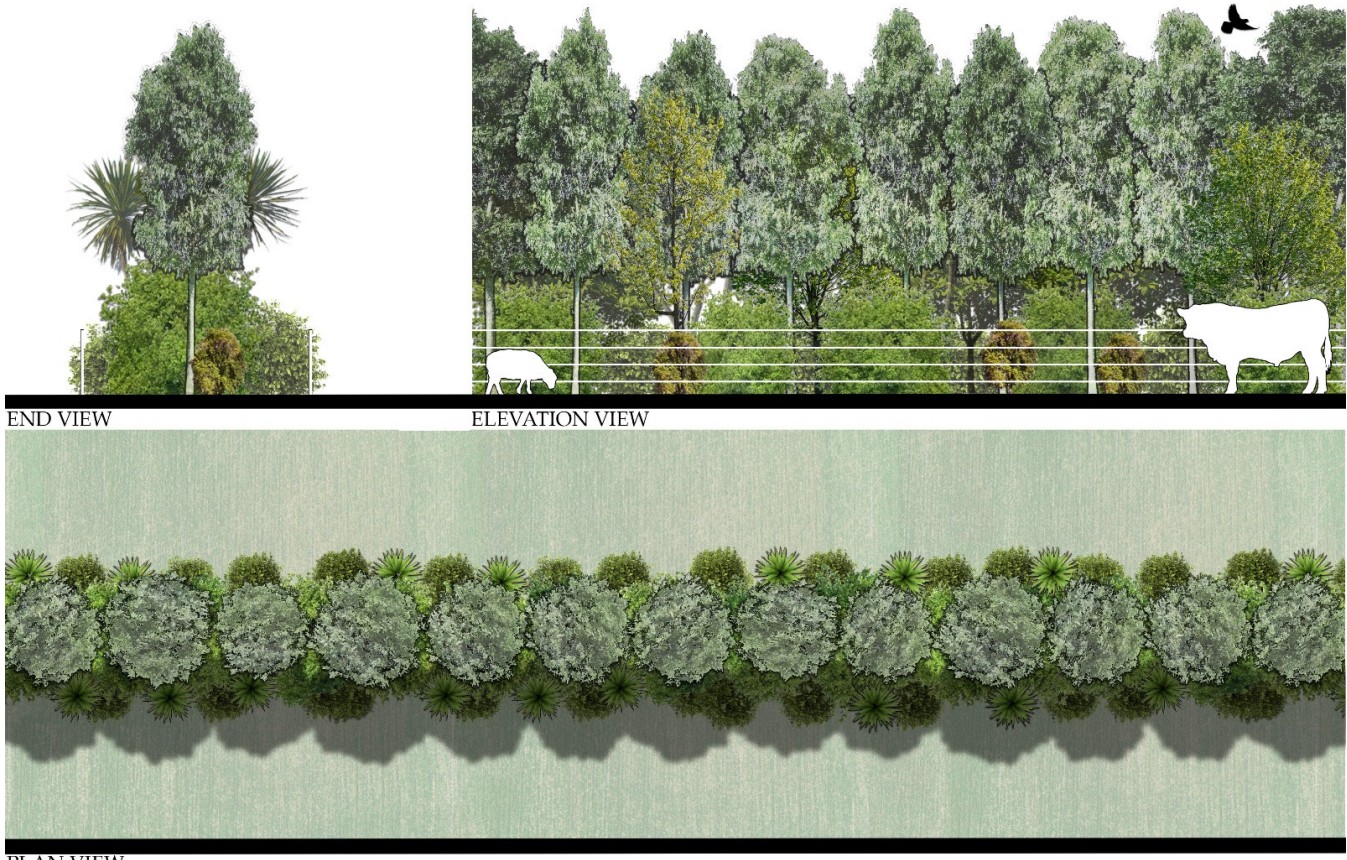

**Figure 8.** End and elevation views (**top**) and plan view (**bottom**) of the straight shelter configuration with spacings between each species related to their height.

### Meandering Shelter

This configuration was designed to enhance animal welfare by offering shade and shelter during various times of the day and diverse weather conditions. The plant spacing resembled the straight shelter configuration but with increased width to allow the plantings to act as a wildlife corridor and provide an edge habitat for native fauna. However, this design choice reduces the available area for pasture grazing (see Figure 9).

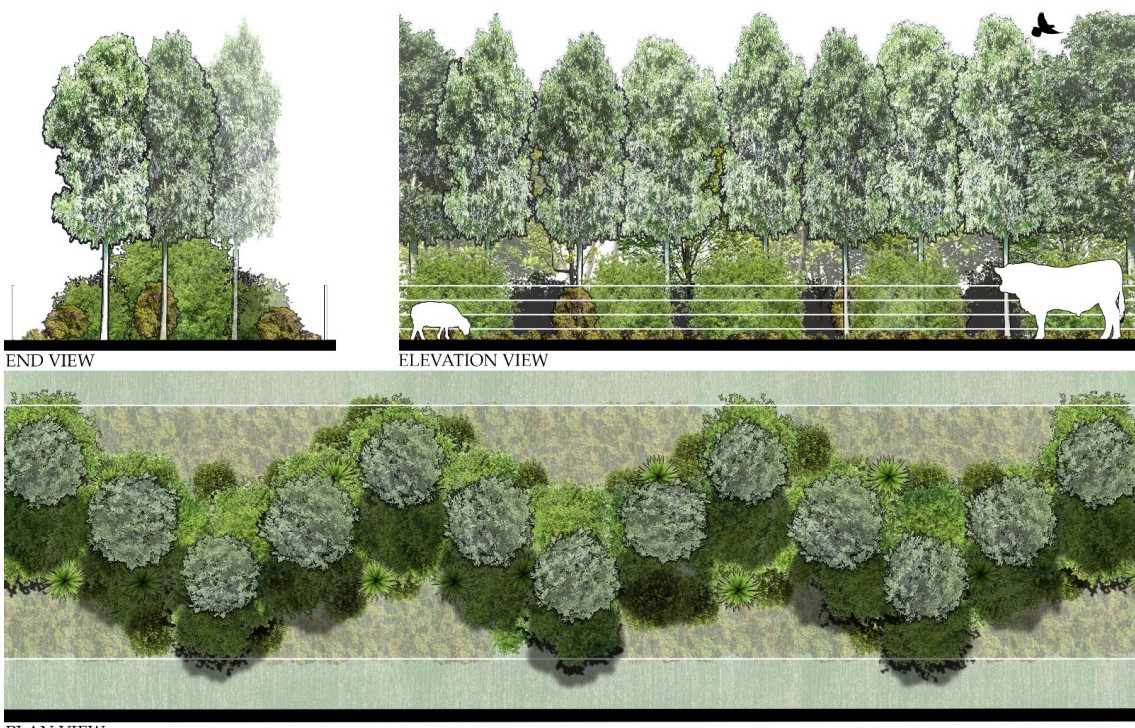

**Figure 9.** End and elevation views (**top**) and plan view (**bottom**) of the meandering shelter configuration.

In-Paddock Shade Trees

This configuration was designed to provide shade throughout the farm by locating taller trees with larger canopies throughout the paddocks, either in groups or as individuals. Shade from these trees follows the sun's arc throughout the day, encouraging livestock to move to areas where the shade is present. The absence of underplanting also allows maximum air movement, contributing to the cooling effect (see Figure 10).

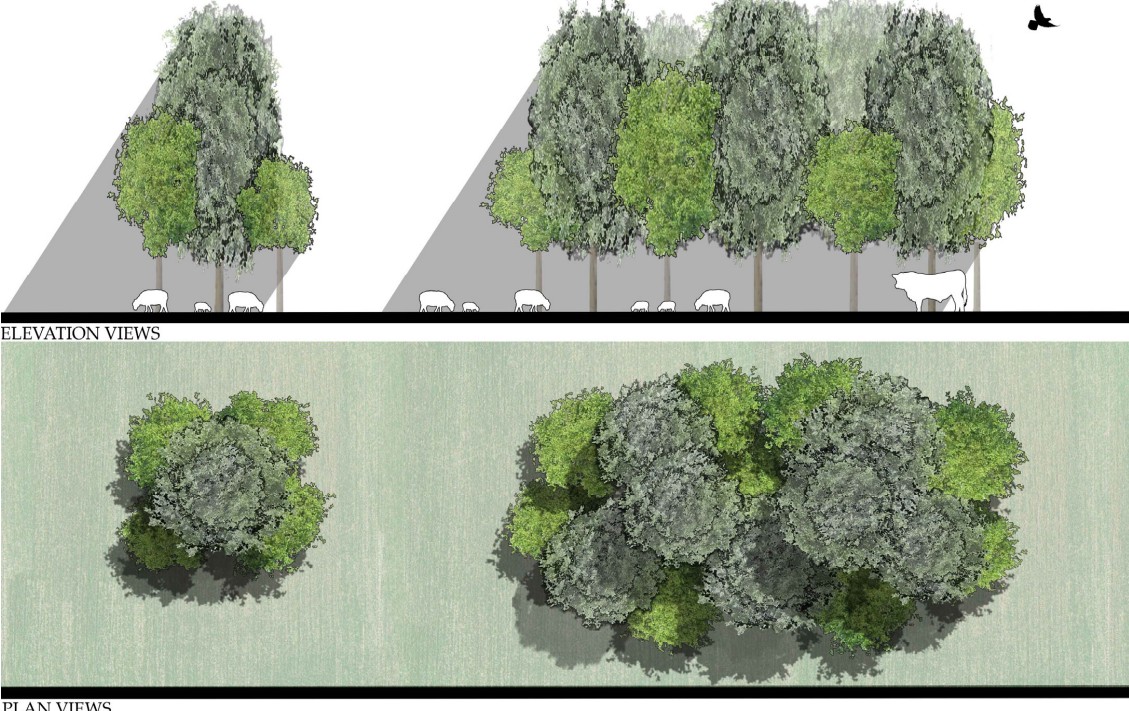

**Figure 10.** Examples of elevation views (**top**) and plan views (**bottom**) of in-paddock shade trees.

### 3.2.4. Shade and Shelter Site Design

Straight shelter configurations are positioned widely across the site, protected from livestock by permanent fence lines and oriented to shield from the three main winds: northeast, hot northwest, and cold southwest. It is predominantly located in the northwest of the Cemetery Block, where precise land areas are necessary for research purposes and in areas where space is limited.

Meandering shelter configurations are situated primarily in two areas with fewer restrictions on space: the south and west of the Homestead Block and beneath the powerline corridor in the Cemetery Block, where lower-growing species are oriented east to west.

In-paddock shade trees are widely distributed throughout the site, aiming to maximize shade coverage in each paddock to mitigate heat stress in livestock [37]. As soils on the site range from deep to shallow and well-drained to poorly drained, tree selection aligns with the soil conditions in which they are planted (see Figure 11).

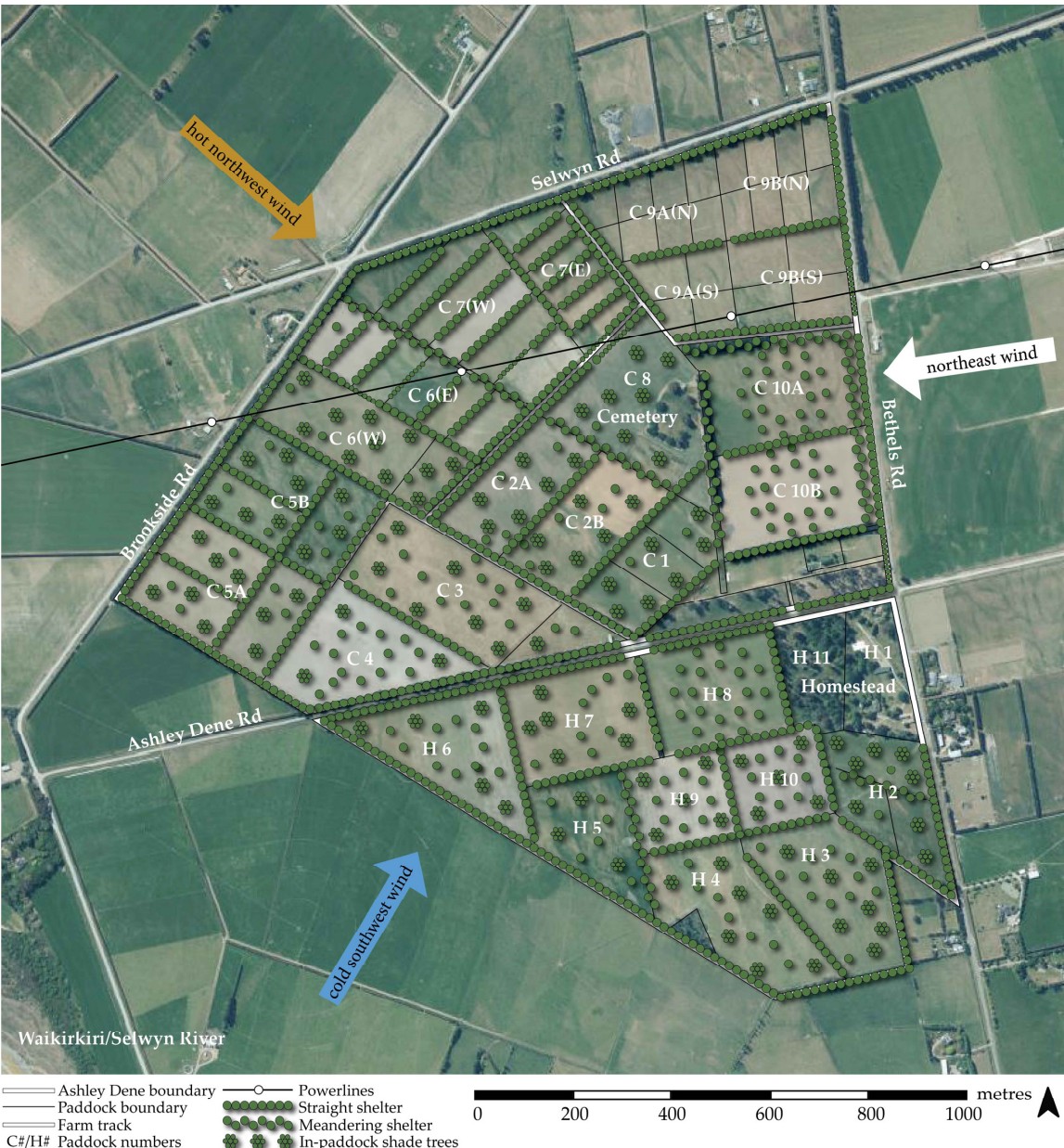

**Figure 11.** Final shade and shelter design for the case study site, with proposed shade and shelter vegetation located to provide shade throughout the day. Arrows indicate the three significant winds experienced at the case study site.

3.2.5. Designing for Native Bush Bird Habitat

The bush bird habitat configurations focus on three areas within the primary habitat patch. These areas are the "existing poplar trees", a "wetland/drainage corridor" and a "densely planted" area.

The key factors for determining the location of the primary 6.5 ha patch on the site are its proximity to two established habitat restoration sites, both within a travel distance of 5 km to 10 km for omnivorous bush birds and 10 km to 25 km for herbivorous bush birds. Additionally, the patch is less than 1 km away from the vegetated Waikirikiri/Selwyn River corridor, which provides a variety of food sources for all bush birds, including insectivorous species [64,73]. Distances between and sizes of the patches and corridors are determined using information from Meurk and Hall (2006) [64].

Habitat Planting Configuration Focused on Poplar Trees

This configuration focuses on the area surrounding the existing poplar trees. Standing poplar trees act as a nurse species for new plantings and fallen poplar trees provide an open canopy for sunlight to enter, enhancing plant growth. Fallen poplar trees also provide habitat for various insects, attracting insectivorous bush birds (see Figure 12).

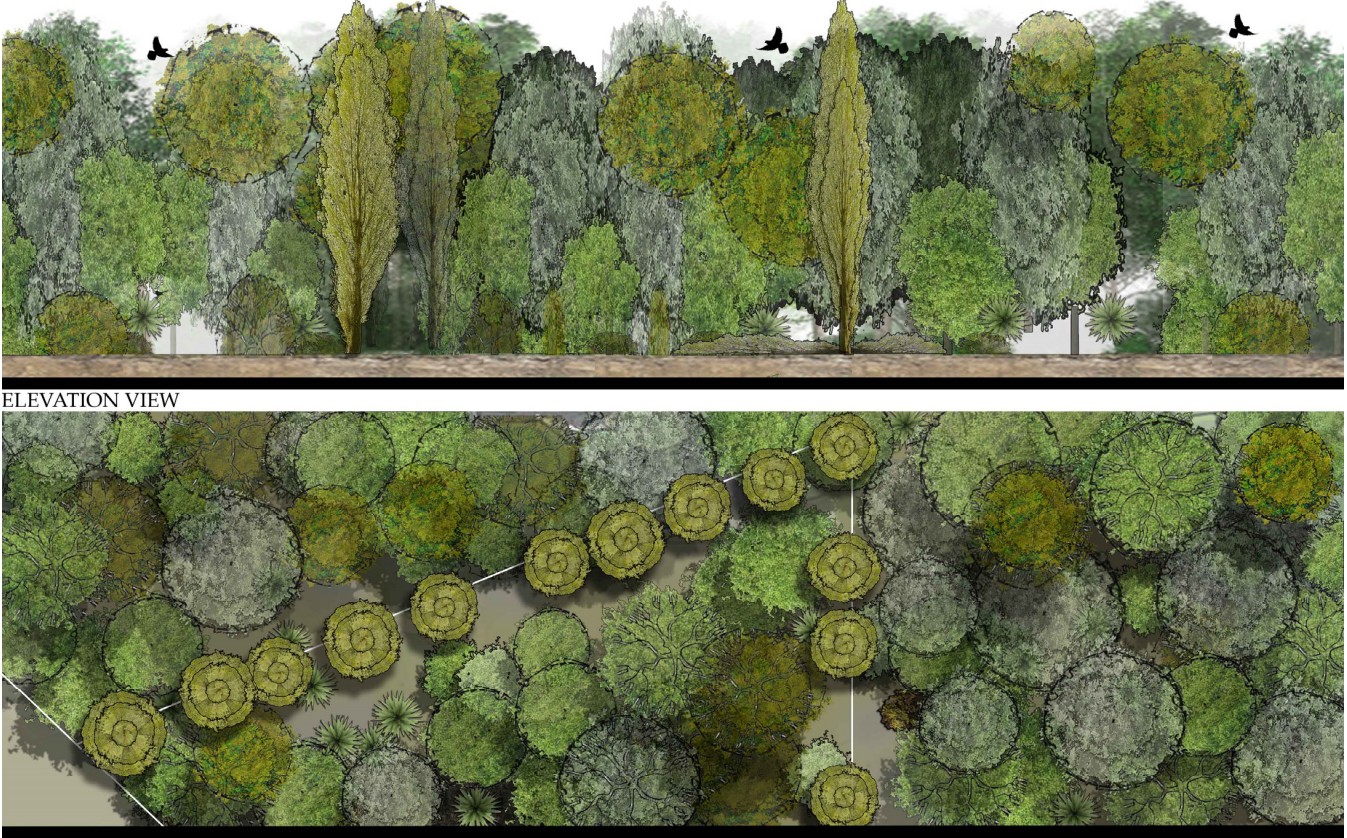

ELEVATION VIEW

PLAN VIEW

**Figure 12.** Elevation view (**top**) and plan view (**bottom**) of habitat area focused on poplar trees.

Habitat Planting Configuration Focused on Wetland/Drainage Corridor

A wetland/drainage corridor in the lowest area of the site is designed to store/filter excess agricultural surface water run-off before it moves into the downstream catchment. Along the existing wetland/drainage channels are lower-growing shrubs that prefer damp soils, surrounded by taller growing species that provide shade to enhance water retention. Waterborne insects should be attracted to this area (See Figure 13).

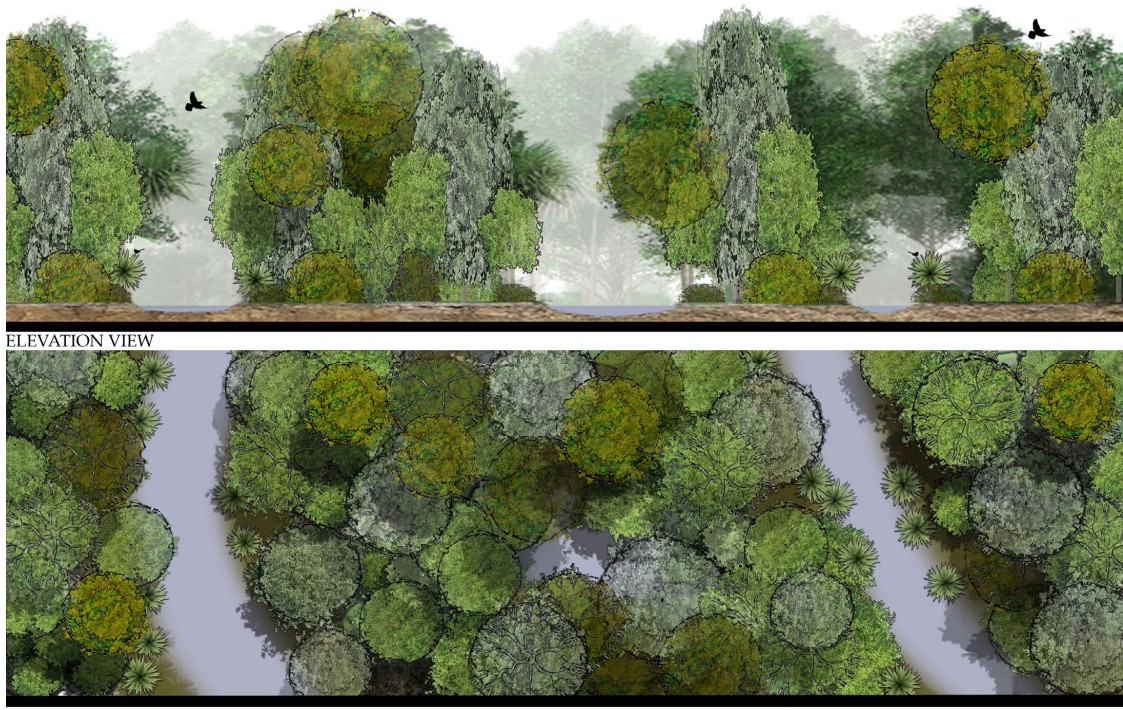

ELEVATION VIEW

PLAN VIEW

**Figure 13.** Elevation view (**top**) and plan view (**bottom**) of habitat area focused on wetland/drainage corridor.

Habitat Planting Configuration Focused on Densely Planted Area

An area of the habitat patch with dense plantings creates an area less susceptible to edge effects. The denser plantings will reduce the distance that light and sound pollution travel into the patch, increasing suitable habitat for more sensitive bush bird species, such as New Zealand pigeon/kererū (*H. novaeseelandiae*) (See Figure 14).

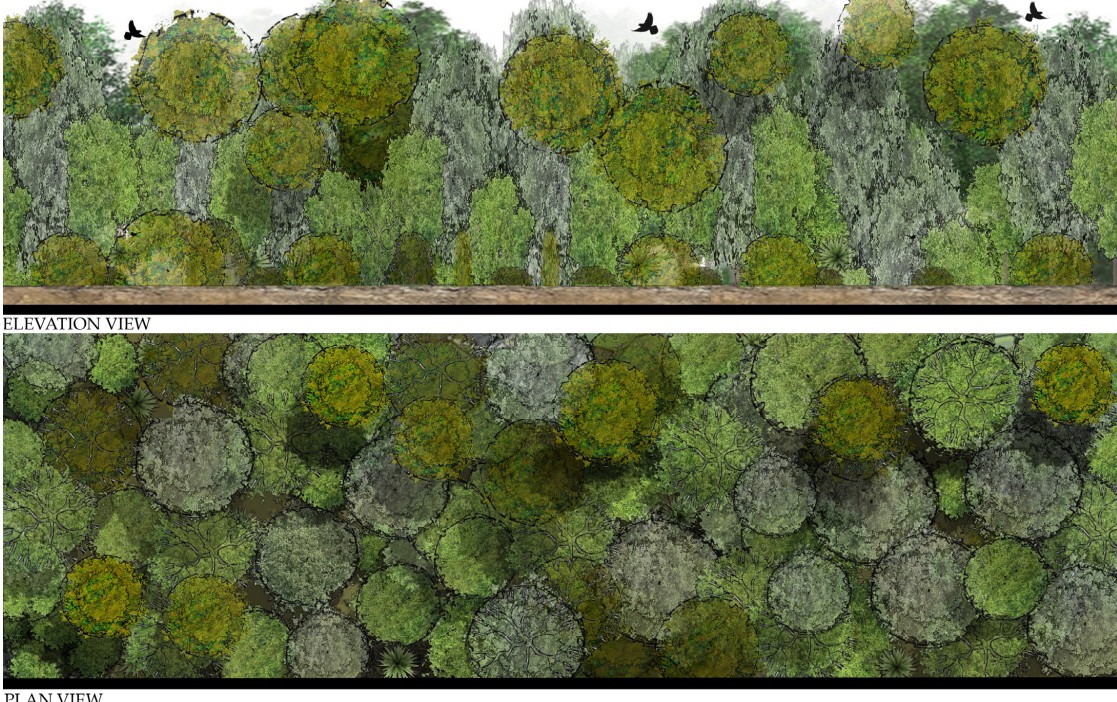

ELEVATION VIEW

PLAN VIEW

**Figure 14.** Elevation view (**top**) and plan view (**bottom**) of habitat area focused on a densely planted area.

### 3.2.6. Bush Bird Habitat Site Design

A primary 6.5 ha habitat patch, comprising a 1.8 ha core and 4.45 ha edge habitat, is situated in the south of the site, covering approximately 4.2% of the overall farm [65,66]. One new physical fence is required across paddocks H3 and H4 to protect the plantings from livestock access. However, no new fences are needed for the secondary 2 ha patch in C5A and the tertiary 1.6 ha patch in C7W. The shelter vegetation is also arranged to create wildlife corridors, enhancing edge habitat area and species diversity [40,41]. Recent studies indicate native bird species richness may increase ca. 122% from an unrestored agricultural landscape, while native bird abundance may increase ca. 272% [74]. While establishing these patches removes 10.1 ha of existing pasture from the 155 ha farm, there will likely be minimal impact on the existing farming operation (see Figure 15).

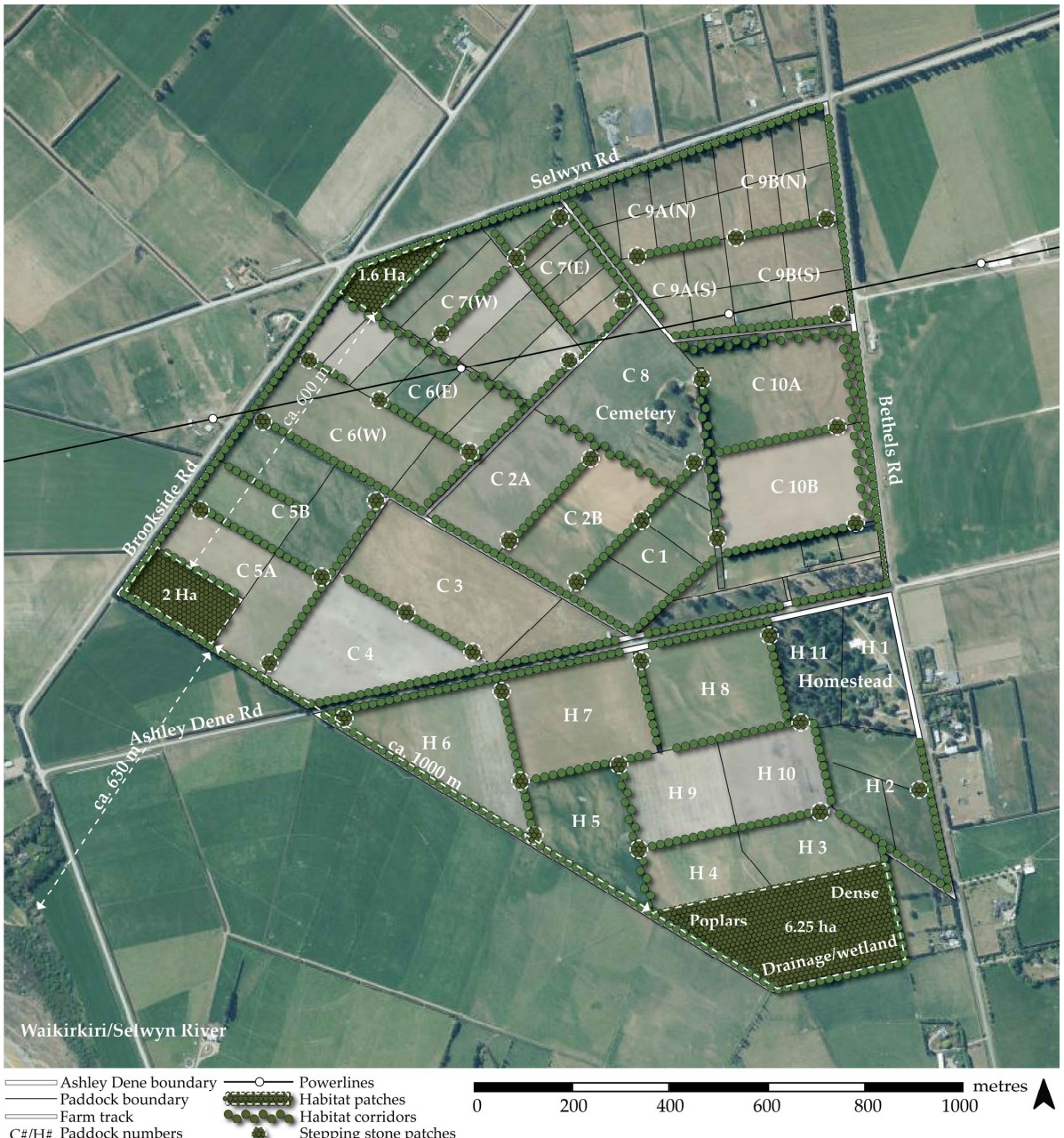

**Figure 15.** Native bush bird habitat site design for the case study site, with proposed habitat patches and corridors.

### 3.3. Combined Multifunctional Site Design

Figure 16 combines the previous site designs of (1) enhancing foraging opportunities, (2) increasing shade and shelter, and (3) restoring native bush bird habitat into one multifunctional design.

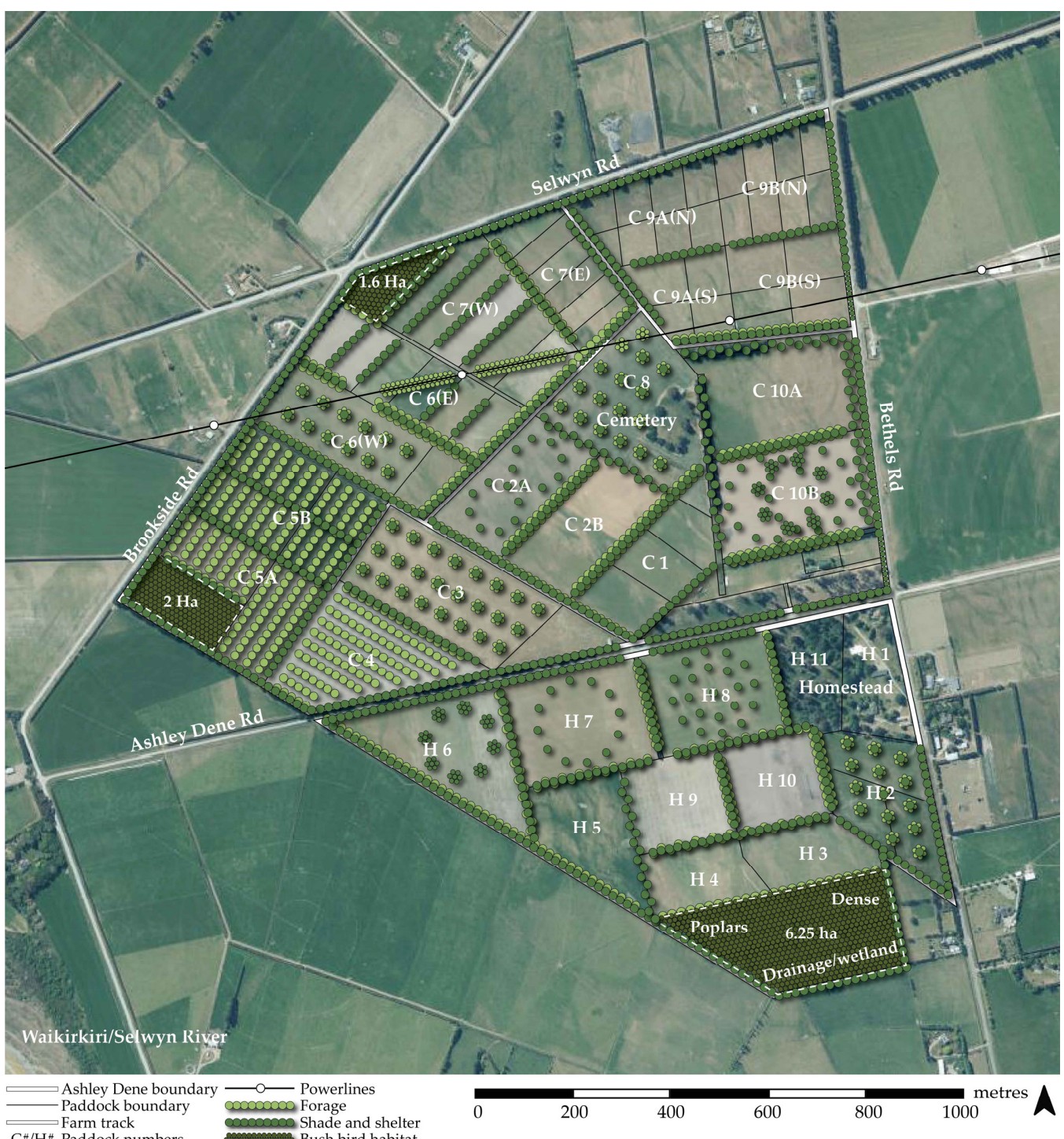

**Figure 16.** A combined multifunctional site design for the entire case study site.

## 4. Discussion

This research aimed to develop a methodology for selecting suitable native woody vegetation in dryland agricultural landscapes and propose spatial configurations and site

designs for its integration on the case study site of Ashley Dene dryland farm. The goal was to benefit livestock, native bush birds, and the wider ecosystem, transitioning the site along the wilderness continuum [75] toward a multifunctional agroecosystem that achieves agricultural and ecological objectives in the same location.

The proposed multifunctional design will significantly increase woody vegetation on the farm from 5.5 ha (3.5%) to 31 ha (20%) of the 155 ha site, reducing the farm's carrying capacity, especially during the summer when water availability is limited. This could be offset by the increase in forage vegetation combined with increased fallen organic matter, which may improve soil health and productivity. Another option is to adjust the stocking policy by selling more young livestock (lambs) earlier, either as store lambs or by changing the breed to sell a higher proportion at weaning. This would lower the stocking rate and pasture demand over the summer. To achieve a 20% reduction in feed demand (from converting 20% of the pasture to woody vegetation), around 40% of the lambs could be sold at weaning, or a more conservative approach of a 50% reduction could be considered as a buffer for summer conditions (A. Greer, personal communication, 25 June 2023).

To optimize the benefits of integrating native vegetation, the study focused on the top 20 species for forage, shade and shelter, and bush bird habitat. However, this approach may limit biodiversity, reducing plant species richness and nutrient availability for livestock. It may also limit the ability of other native fauna, such as invertebrates and lizards, from returning to the landscape as their preferred food sources and habitat conditions may not be present. Incorporating environmental and soil scores in the multifunctional site design improves the accuracy of plant selection and is recommended for future studies where the site has greater variation in soil and environmental conditions.

### 4.1. Native Woody Vegetation and Spatial Layouts for Enhanced Foraging Opportunities

The most suitable species for forage are shrubs and small trees, as they are lower to the ground and easier to access for sheep and cattle. The most suitable species include makomako, wineberry (*Aristotelia serrata*), karamu (*Coprosma robusta*), teucridium (*Teucridium parvifolium*), lemonwood, tarata (*Pittosporum eugenioides*), black matipo, kōhūhū (*Pittosporum tenuifolium*) and broadleaf, kāpuka (*Griselinia littoralis*), (see Table 2). The low height of these species allows them to be placed in areas with height restrictions, maximizing utilization beneath power lines, pivot irrigators, and flight paths.

Twenty-five paddocks are dedicated to grazing, with improved shelter vegetation within approximately 80 m of any point on the farm to enhance animal welfare [36,37] (distances measured from Figure 16). The northeast area of the Cemetery Block is reserved for pasture and livestock research trials, with additional forage strategically placed in sheltered paddocks for the lambing season. Forage strips are positioned 40 m apart to align with optimal livestock browsing patterns [33]. Furthermore, the orientation of forage plantings maximizes sun exposure and provides shelter from the three most significant winds on-site. While livestock benefits from a diverse diet, providing a range of nutrients and self-medication possibilities, further research is required on the nutritional value of native woody vegetation in New Zealand.

### 4.2. Native Woody Vegetation and Spatial Layouts for Optimal Shade and Shelter On-Farm

The most suitable species to provide shade and shelter include taller trees such as tōtara (*Podocarpus totara*), matai (*Prumnopitys taxifolia*), and kahikatea (*Dacrycarpus dacrydioides*) and denser trees and shrubs, such as pōkākā (*Elaeocarpus hookerianus*), lemonwood (*Pittosporum eugenioides*) and narrow-leaved houhere (*Hoheria angustifolia*), (see Table 3).

Increasing shade and shelter has implications for soil conditions, with shaded areas having lower temperatures year-round. This can benefit animal welfare in warmer months but may prolong frosts and affect pasture growth and soil moisture retention in cooler months, posing challenges for farm usability. Therefore, integrating exotic and native species in dryland agricultural landscapes requires further investigation.

Providing livestock access to shade throughout the site and protection from significant winds, including hot, dry northwest winds, cold, wet southwest winds, and prevailing cool northeast winds, is critical for livestock welfare. Increased shaded and sheltered areas can mitigate heat stress, prevent excess moisture loss, improve animal well-being, and enhance lamb survival rates.

While individual trees pruned into a lollipop shape are effective for shade, livestock movement is necessary to prevent soil compaction or pugging, which are detrimental to animal welfare and nearby vegetation. However, this can pose challenges for using conventional farm machinery when it is required to move around the trees. Shelterbelts may be less effective under humid conditions with low wind speeds (M. Bloomberg, personal communication, 25 July 2022).

Planting shade and shelter along boundaries can impact neighboring properties, potentially affecting soil water retention and wind patterns. Coordination between landowners is crucial in such cases.

### 4.3. Native Woody Vegetation and Spatial Layouts for Optimal Bush Bird Habitat

The most suitable species for enhancing bush bird habitat in dryland agricultural landscapes include five-finger, whauwhaupaku (*Pseudopanax arboreus*), kōwhai, small-leaved kōwhai (*Sophora microphylla*), cabbage tree, ti kōuka (*Cordyline australis*), marbleleaf, putaputāwētā (*Carpodetus serratus*), and kanuka (*Kunzea ericoides*), (see Table 4). However, as it is widely accepted that native bush birds are known to nest in large exotic trees, often travelling considerable distances in search of food, more plant species could easily be included in this list.

The site designs for forage, shade and shelter focused on the site inventory, while the bush bird habitat designs considered vegetation in the wider landscape, specifically the primary 6.25 ha habitat patch, which is located within 5 km of existing habitat restoration sites in the landscape and must be connected to the wider ecological network for bush bird species to repopulate the landscape [64]. Secondary and tertiary habitat patches are established on the farm, assuming the livestock diet will be enhanced by increased forage availability and improved pasture quality resulting from higher organic matter deposition from additional woody vegetation. The smaller 0.01 ha habitat patches were removed as the shelterbelts were deemed to provide sufficient edge habitat. Removing these areas can decrease the *quantity* of available food sources [76]; however, this can be compensated for by improving the *quality* of the food sources, through effective plant selection.

Restoring and integrating habitat areas may encourage other landowners to establish habitat areas and support the return of native bush birds to the fragmented landscape.

### 4.4. Native Woody Vegetation and Spatial Layouts for Optimal Functions When Considering the Three Components of a Multifunctional Dryland Agricultural Landscape

When considering the three multifunctional components within this research, the most suitable species for providing multifunctional benefits in farming systems include tōtara (*Podocarpus totara*), narrow-leaved houhere (*Hoheria angustifolia*), pōkākā (*Elaeocarpus hookerianus*), lowland ribbonwood (*Plagianthus regius*), matai (*Prumnopitys taxifolia*), cabbage tree, ti kōuka (*Cordyline australis*), black matipo (*Pittosporum tenuifolium*), broadleaf, kāpuka (*Griselinia littoralis*), and kanuka (*Kunzea ericoides*) are the most likely species of native woody vegetation to provide optimal multifunctional benefits. Incorporating these species enhances forage, shade and shelter, and bush bird habitat, improving ecosystem services compared to monoculture systems.

Forage plantings on the case study site are strategically placed adjacent to the habitat patches, creating multifunctional forage/habitat areas. Integrating forage configurations with shade and shelter configurations also forms multifunctional forage/shade areas, which have been shown to increase forage consumption while maintaining the maximum area for pasture grazing.

A site inventory and analysis should be carried out before designing the interventions. Although the principles of establishing shade and shelter can be adapted to various farming systems, it is important to customize these interventions according to each site's specific soil and climatic conditions to optimize the design's effectiveness. Applying this research to other farming systems in different locations, with different bioclimatic conditions, vegetation, and livestock types, will contribute valuable information to this topic.

## 5. Conclusions

The study highlights that integrating native woody vegetation into agricultural systems can positively affect livestock welfare and restore native bush birds in fragmented landscapes. By restoring native bush bird habitat, it becomes possible to develop a financially sustainable farming model that restores degraded agricultural landscapes and endemic biodiversity and strengthens the national identity of Aotearoa New Zealand. Furthermore, applying this model to other regions, locally, regionally, nationally, and internationally, has the potential to amplify these benefits.

**Author Contributions:** Methodology, J.E., S.D. and C.D.; software, J.E.; validation, J.E., S.D., C.D. and P.G.; formal analysis, J.E.; investigation, J.E.; resources, J.E.; data curation, J.E.; writing—original draft preparation, J.E., S.D. and C.D.; writing—review and editing, J.E., S.D., C.D. and P.G.; visualization, J.E.; supervision, S.D. and C.D.; project administration, J.E.; funding acquisition, J.E. All authors have read and agreed to the published version of the manuscript.

**Funding:** This research received no external funding.

**Institutional Review Board Statement:** Not applicable.

**Informed Consent Statement:** Not applicable.

**Data Availability Statement:** Not applicable.

**Acknowledgments:** J.E. would like to acknowledge the following individuals: Colin Meurk for his assistance with plant information and ecology; Catherine Eggers for ongoing support.

**Conflicts of Interest:** The authors declare no conflict of interest.

## Appendix A

**Table A1.** Multifunctional native plant list for dryland agroecosystems (weighted scores)—groupings.

| Botanical Name | Common Name | Fo | SS | HT | ET | So | Total |
| --- | --- | --- | --- | --- | --- | --- | --- |
| *Podocarpus totara* | tōtara | 5.00 | 25.00 | 6.00 | 22.50 | 20.75 | 79.25 |
| *Hoheria angustifolia* | narrow-leaved houhere | 10.00 | 20.50 | 6.50 | 21.00 | 20.75 | 78.75 |
| *Elaeocarpus hookerianus* | pōkākā | 8.00 | 22.50 | 6.50 | 17.50 | 21.75 | 76.25 |
| *Plagianthus regius* | lowland ribbonwood, mānatu | 8.00 | 20.50 | 6.00 | 19.50 | 20.75 | 74.75 |
| *Prumnopitys taxifolia* | mataī, black pine | 5.00 | 25.00 | 6.50 | 15.50 | 20.75 | 72.75 |
| *Cordyline australis* | cabbage tree, ti kōuka | 11.00 | 15.00 | 7.00 | 21.50 | 17.75 | 72.25 |
| *Pittosporum tenuifolium* | black matipo, kōhūhū | 11.00 | 18.00 | 5.00 | 19.00 | 18.50 | 71.50 |
| *Griselinia littoralis* | broadleaf, kāpuka | 11.00 | 17.50 | 5.50 | 20.50 | 17.00 | 71.50 |
| *Kunzea ericoides* | kānuka | 7.00 | 13.50 | 7.00 | 24.00 | 20.00 | 71.50 |
| *Dodonaea viscosa* | akeake | 9.00 | 18.00 | 4.50 | 19.50 | 18.75 | 69.75 |
| *Pittosporum eugenioides* | lemonwood, tarata | 11.00 | 21.50 | 5.00 | 14.50 | 15.50 | 67.50 |
| *Dacrycarpus dacrydioides* | kahikatea | 5.00 | 24.00 | 6.50 | 14.50 | 17.25 | 67.25 |
| *Olearia fragrantissima* | fragrant tree daisy | 9.00 | 14.00 | 4.50 | 18.00 | 20.75 | 66.25 |
| *Corynocarpus laevigatus* | karaka | 8.00 | 17.50 | 5.00 | 15.50 | 19.25 | 65.25 |

**Table A1.** *Cont.*

| Botanical Name | Common Name | Fo | SS | HT | ET | So | Total |
|---|---|---|---|---|---|---|---|
| *Olearia avicenniifolia* | mountain akeake | 9.00 | 10.00 | 4.50 | 22.50 | 18.50 | 64.50 |
| *Pseudopanax arboreus* | five-finger, whauwhaupaku | 11.00 | 15.00 | 7.50 | 14.50 | 16.25 | 64.25 |
| *Olearia paniculata* | akiraho, golden akeake | 9.00 | 12.50 | 4.50 | 19.00 | 19.25 | 64.25 |
| *Sophora microphylla* | kōwhai, small-leaved kōwhai | 6.00 | 11.00 | 6.50 | 21.50 | 18.50 | 63.50 |
| *Discaria toumatou* | matagouri, tūmatakuru | 9.00 | 10.00 | 2.00 | 23.50 | 19.00 | 63.50 |
| *Leptospermum scoparium* | mānuka | 7.00 | 12.50 | 4.00 | 23.50 | 16.25 | 63.25 |
| *Pseudopanax crassifolius* | lancewood, horoeka | 9.00 | 13.00 | 3.50 | 16.50 | 20.75 | 62.75 |
| *Coprosma propinqua* | mikimiki | 10.00 | 10.00 | 2.00 | 22.50 | 17.75 | 62.25 |
| *Muehlenbeckia astonii* | shrubby tororaro | 8.00 | 10.00 | 2.50 | 24.00 | 17.75 | 62.25 |
| *Corokia cotoneaster* | korokio, wire-netting bush | 10.00 | 9.00 | 2.50 | 22.50 | 17.75 | 61.75 |
| *Teucridium parvifolium* | teucridium | 12.00 | 6.00 | 2.00 | 22.50 | 19.00 | 61.50 |
| *Pseudopanax ferox* | fierce lancewood | 6.00 | 11.00 | 3.50 | 21.75 | 19.25 | 61.50 |
| *Coprosma virescens* | mikimiki | 11.00 | 7.50 | 2.00 | 22.50 | 18.50 | 61.50 |
| *Olearia adenocarpa* | Canterbury shrub daisy | 7.00 | 7.50 | 2.50 | 24.50 | 20.00 | 61.50 |
| *Helichrysum lanceolatum* | niniao | 9.00 | 7.50 | 2.50 | 22.50 | 18.50 | 60.00 |
| *Carpodetus serratus* | marbleleaf, putaputāwētā | 10.00 | 13.00 | 7.00 | 19.00 | 11.00 | 60.00 |
| *Carmichaelia australis* | common broom | 9.00 | 6.00 | 2.50 | 23.50 | 18.75 | 59.75 |
| *Muehlenbeckia complexa* | small-leaved pōhuehue | 8.00 | 10.00 | 2.00 | 21.50 | 17.75 | 59.25 |
| *Coprosma intertexta* | mikimiki | 10.00 | 7.50 | 2.00 | 21.00 | 18.75 | 59.25 |
| *Olearia lineata* | shrub daisy | 9.00 | 10.00 | 2.50 | 20.00 | 17.00 | 58.50 |
| *Ozothamnus leptophyllus* | tauhinu | 8.00 | 7.50 | 1.50 | 22.50 | 18.75 | 58.25 |
| *Veronica strictissima* | Banks Peninsula hebe | 11.00 | 8.50 | 2.00 | 18.75 | 17.75 | 58.00 |
| *Veronica salicifolia* | koromiko | 11.00 | 11.50 | 2.00 | 15.50 | 17.75 | 57.75 |
| *Carmichaelia torulosa* | Canterbury broom | 10.00 | 5.50 | 2.50 | 21.00 | 18.75 | 57.75 |
| *Coprosma lucida* | karangu, shining karamū | 11.00 | 11.00 | 2.00 | 15.50 | 17.75 | 57.25 |
| *Coprosma linariifolia* | yellow wood | 10.00 | 11.50 | 2.00 | 16.00 | 17.25 | 56.75 |
| *Coprosma robusta* | karamū | 12.00 | 11.00 | 2.00 | 14.50 | 17.00 | 56.50 |
| *Aristotelia fruticosa* | mountain wineberry | 11.00 | 4.50 | 2.50 | 20.50 | 17.75 | 56.25 |
| *Sophora prostrata* | prostrate kōwhai | 7.00 | 4.50 | 2.50 | 22.50 | 19.00 | 55.50 |
| *Coprosma acerosa* | sand coprosma | 8.00 | 6.00 | 2.00 | 21.00 | 18.50 | 55.50 |
| *Pennantia corymbosa* | kaikōmako | 10.00 | 14.50 | 2.00 | 17.50 | 11.25 | 55.25 |
| *Phormium tenax* | flax, harakeke | 9.00 | 7.50 | 2.50 | 19.50 | 16.25 | 54.75 |
| *Melicytus alpinus* | porcupine shrub | 6.00 | 5.00 | 2.50 | 22.50 | 18.50 | 54.50 |
| *Coprosma crassifolia* | mikimiki | 10.00 | 7.50 | 2.00 | 22.50 | 12.50 | 54.50 |
| *Plagianthus divaricatus* | marsh ribbonwood | 8.00 | 7.00 | 2.50 | 20.00 | 16.25 | 53.75 |
| *Olearia bullata* | shrub daisy | 10.00 | 10.50 | 2.50 | 19.00 | 11.75 | 53.75 |
| *Muehlenbeckia axillaris* | creeping pōhuehue | 7.00 | 6.00 | 2.00 | 21.00 | 16.25 | 52.25 |
| *Aristotelia serrata* | makomako, wineberry | 12.00 | 14.50 | 4.50 | 9.50 | 11.25 | 51.75 |
| *Myrsine divaricata* | weeping matipo, māpou | 8.00 | 9.50 | 2.50 | 16.00 | 14.75 | 50.75 |
| *Coprosma wallii* | mikimiki | 10.00 | 7.50 | 2.00 | 18.00 | 11.75 | 49.25 |
| *Coprosma dumosa* | coprosma | 10.00 | 3.50 | 2.00 | 22.50 | 11.00 | 49.00 |

| Botanical Name | Common Name | Fo | SS | HT | ET | So | Total |
|---|---|---|---|---|---|---|---|
| *Melicope simplex* | poataniwha | 5.00 | 10.00 | 2.00 | 17.50 | 13.25 | 47.75 |
| *Chionocloa conspicua* | hunagamoho | 9.00 | 3.50 | 1.50 | 22.50 | 11.00 | 47.50 |
| *Austroderia richardii* | toetoe | 6.00 | 9.00 | 2.00 | 19.50 | 10.25 | 46.75 |
| *Cyathodes juniperina* | mingimingi | 5.00 | 6.00 | 2.00 | 20.50 | 11.75 | 45.25 |
| *Coprosma rigida* | rigid mikimiki | 9.00 | 6.50 | 2.00 | 14.50 | 13.00 | 45.00 |
| *Coprosma pedicellata* | mikimiki | 10.00 | 11.50 | 2.00 | 15.00 | 6.50 | 45.00 |
| *Anemanthele lessoniana* | wind grass | 9.00 | 1.00 | 1.50 | 18.50 | 14.75 | 44.75 |
| *Astelia fragrans* | bush lily | 9.00 | 2.00 | 2.00 | 11.50 | 11.25 | 35.75 |

Key: Fo: Forage; SS: Shade/Shelter; HT: Habitat; ET: Environmental tolerances; So: Soil preferences.

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
