# Peer review of "Enhancing Multifunctionality in Agricultural Landscapes with Native Woody Vegetation"

_sustainability, doi:10.3390/su151411295_

Round 1

Reviewer 1 Report

*This manuscript review the potential of integrating native woody vegetation for increased multifunctionality within dryland agricultural landscapes. 

*I think the research topic will be interesting for the readers of the Journal. 

*The paper is well written and easy to understand.

*The abstract is very clearly described and comprehensive.

*The English needs to be revised in all sections so that the content is written more succinctly, coherently and logically, avoiding unnecessary repetition.

*Check the location of references throughout the manuscript. For example, lines 90, incorrect reference number.

*In addition, there is an inaccuracy in the ordering of the references. (From 64 to 69). The ordering must be done correctly.

*Check the format of each reference.

Author Response

Thank you for taking the time to review this manuscript and provide your valuable feedback. I have attached the updated version. Kind regards. James Eggers.

Reviewer 2 Report

It is a very good try to integrate the native woody vegetation into agricultural systems. As the authors mentioned in the manuscript the reintegration of native woody vegetation within agricultural landscapes can benefits for the environment, livestock and wildlife. It’s a good story. However, I still have some questions:

1.     What was the state of native vegetation before reclamation? Could you make a comparison with the agricultural landscapes?

2.     Please give us some basic information about native bush birds. Again, is it possible to give a comparison before and after reclamation?

3.     Please provide basic information about the livestock on the farm.

4.     Have you distinguished the forage preferences of different livestock?

5.     Did you consider the relationships between your design and the carrying capacity of the farm?

6.     L186   What species is the palatability of forage mainly directed to?

7.     L202 What is the criteria that these weightings were assigned to each characteristic?

8.     L257 Does CB in Figure3 and the other Figures mean C8? 

9.     L388   error

10.  L447   error

11.  L454 How many native bush birds can your habitat design support?

12.  L559  For what species?

13.  Please raise the resolution of your Figures.

Author Response

Thank you for taking the time to review this manuscript and provide valuable feedback. I have attached the updated version. Kind regards, James Eggers.

Reviewer 3 Report

There are some comments, which are to be incorporated in order to improve the manuscript, as given below:

*The whole manuscript should be formatted properly according to the journal style.

*Abstract is not written well. Author must reconstruct the abstract of the paper.

*Need of the work is not well formulated in the ‘Introduction’ section. The authors did not present a novel justification for carrying out this work.

*Introduction section needs more improvement.

*The novelty of the work must be identified and stated more carefully. The authors have to try to explain why this paper is relevant to the wider readership.

*Authors should show the limitations of previous papers.

*Image quality must be improved.

*Overall the manuscript needs more improvement. It is too superficial and not a meaningful writing.

*Authors need to rephrase the "Conclusions" section and write it in more meaningfully.

Author Response

(The authors gave the same response as above.)

Round 2

Reviewer 2 Report

I am happy to see the revised version of the manuscript. It has been improved greatly. 

Reviewer 3 Report

I recommended that the manuscript be accepted in its current form. The authors have made major improvements to the paper in response to the comments made.